# Reconstructing Cell Lineage Trees from Phenotypic Features with Metric Learning

**Da Kuang** [1]    **Guanwen Qiu** [1]    **Junhyong Kim** [1] [2]

## Abstract

How a single fertilized cell gives rise to a complex array of specialized cell types in development is a central question in biology. The cells replicate to generate cell lineages and acquire differentiated characteristics through poorly understood molecular processes. A key approach to studying developmental processes is to infer the tree graph of cell lineage histories, which provides an analytical framework for dissecting individual cells' molecular decisions during replication and differentiation (i.e., acquisition of specialized traits). Although genetically engineered lineage-tracing methods have advanced the field, they are either infeasible or ethically constrained in many organisms. By contrast, modern single-cell technologies can measure high-content molecular profiles (*e.g.*, transcriptomes) in a wide range of biological systems. Here, we introduce *CellTreeQM*, a novel deep learning method based on transformer architectures that learns an embedding space with geometric properties optimized for tree-graph inference. By formulating the lineage reconstruction problem as tree-metric learning, we systematically explore weakly supervised training settings at different levels of information and present the *Cell Lineage Reconstruction Benchmark* to facilitate comprehensive evaluation. This benchmark includes (1) synthetic data modeled via Brownian motion with independent noise and spurious signals; (2) lineage-resolved single-cell RNA sequencing datasets. Experimental results show that *CellTreeQM* recovers lineage structures with minimal supervision and limited data, offering a scalable framework

for uncovering cell lineage relationships. To our knowledge, this is the first method to cast cell lineage inference explicitly as a metric learning task, paving the way for future computational models aimed at uncovering the molecular dynamics of cell lineage. Code and benchmarks are available at: https://kuang-da.github.io/CellTreeQM-page

## 1. Introduction

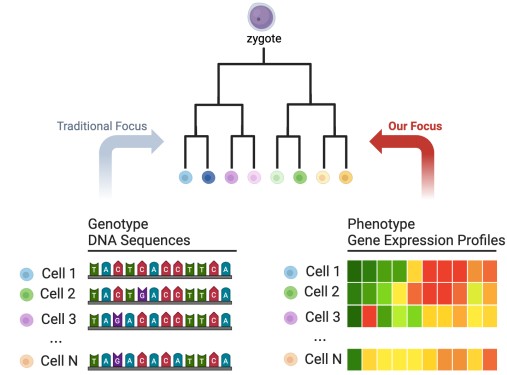

*Figure 1.* **Exploring phenotype-based cell lineage reconstruction.** This figure highlights the focus of our study on reconstructing cell lineage trees using phenotype data, specifically gene expression profiles (right panel), in contrast to traditional methods that rely on genotype data, such as DNA sequences (left panel).

Understanding how a fertilized egg's repeated divisions and differentiation, *i.e.*, the process of different cells acquiring unique characteristics (*e.g.*, skin cell, muscle cell), gives rise to a fully formed embryo has been a long-standing goal in biology (Wolpert et al., 2015; Slack & Dale, 2021). A key component of this developmental process is the tree-graph of *cell lineages*, which provides a roadmap of how diverse cell types arise from a single progenitor (Clevers, 2011; Shapiro et al., 2013; Wagner & Klein, 2020). The cell replication process cannot be directly observed in most organisms; therefore, inferring or reconstructing the cell lineage tree from features measured on the individual cells is an important challenge. Beyond organismal development,

[1]Department of Computer and Information Science, University of Pennsylvania, Philadelphia, USA [2]Department of Biology, University of Pennsylvania, Philadelphia, USA. Correspondence to: Da Kuang <kuangda@seas.upenn.edu>, Junhyong Kim <junhyong@sas.upenn.edu >.

*Proceedings of the $42^{nd}$ International Conference on Machine Learning*, Vancouver, Canada. PMLR 267, 2025. Copyright 2025 by the author(s).

knowledge of cell lineages has wide biomedical applications including deciphering molecular processes underlying cell injury and repair, tumor development, degenerative diseases, etc.(Zhang et al., 2020; Sivandzade & Cucullo, 2021)

Currently, the gold standard for cell lineage tree reconstruction is *prospective lineage tracing* (Kretzschmar & Watt, 2012). One popular approach leverages CRISPR-Cas9 to genetically engineer "recorders"—exogenous DNA sequences that accumulate heritable mutations (Fig. 1, left). Although powerful, these recorders face key limitations. Their mutation capacity is constrained by the size of the target array, and the overall mutation rate may be uninformative. Most importantly, since it involves genetic engineering, it can only be used in contexts where such genome manipulation is feasible or ethical (McKenna & Gagnon, 2019; Zafar et al., 2020).

Meanwhile, before the availability of molecular sequences, there was a rich history of using *phenotypes* to infer lineage relationships via phylogenetic methods. For foundational results and methodologies in this field, we refer readers to (Kim & Warnow, 1999). Today, advancements in single cell biology have enabled high-content molecular phenotype measurements from individual cells. For example, the total RNA content of a cell, called the transcriptome, consisting of a vector of counts of different RNA species, can now be routinely obtained. **An open question is whether such high-content molecular phenotypes contain sufficient information for tree reconstruction, and if so, what algorithm can be used to recover it** (Fig. 1, right).

We show a positive answer to the above question by introducing *CellTreeQM* (Cell-Tree Quartet Metric), a deep learning framework built on transformer architectures that maps transcriptome data to an embedding space where distances reflect tree-like relationships. Our results demonstrate that reconstructing cell lineage structures from transcriptome data alone is both tractable and data-efficient.

Our main contributions are as follows:

**Formulating the Cell Lineage Tree Reconstruction Problem.** We frame the reconstruction of cell lineage trees as a metric learning problem and identify three practical settings: supervised, weakly supervised, and unsupervised (Fig. 2). Within the weakly supervised paradigm, we consider two realistic scenarios. In the *high-level partition setting*, biologists possess prior knowledge of major clades within the lineage. In the *partial-leaf labeled setting*, lineage-tracing technologies provide topological labels for a subset of cells, with the objective of extrapolating these relationships to unlabeled leaves.

**Proposed Solution.** We present *CellTreeQM*, a feature learning framework for cell lineage tree reconstruction. Inspired by (De Soete, 1983), we design a loss function that explicitly encourages the learned embedding space to satisfy tree-metric properties, and show that stochastic gradient descent efficiently identifies generalizable embeddings.

**Lineage Reconstruction Benchmark.** We introduce a Lineage Reconstruction Benchmark comprising (a) synthetic datasets based on Brownian motion with independent noise and spurious signals, (b) lineage-resolved scRNA-seq datasets. Experimental results on the benchmark demonstrate that *CellTreeQM* efficiently reconstructs lineage structures under weak supervision and limited data, providing a scalable framework for uncovering cell lineage relationships.

## 2. Key Challenges

Reconstructing a cell lineage tree from *phenotypic* data conceptually parallels phylogenetics, which infers evolutionary relationships from discrete data such as aligned DNA or protein sequences. Classical phylogenetic methods typically rely on well-defined stochastic models of sequence evolution (e.g., Jukes–Cantor) to estimate both topology and branch lengths (Felsenstein, 2003). In contrast, the use of phenotypic data for lineage reconstruction remains largely unexplored. We attribute this gap primarily to two key challenges, outlined below. These challenges collectively underscore why comprehensive lineage annotations remain scarce and why robust methods for lineage reconstruction—particularly under incomplete or noisy labels—are urgently needed.

**Uncharacterized Stochastic Processes in Gene Expression Data** In phylogenetics, genetic evolution is often modeled using well-defined stochastic processes, which have been empirically validated across diverse datasets. These models provide a principled framework for lineage reconstruction, typically through maximum likelihood estimation.

However, to the best of our knowledge, no well-established lineage-dependent process exists for phenotypic gene expression data. In fact, during cell differentiation in most organisms, cells that acquire particular phenotypic states (i.e. cell types) are often not monophyletic (i.e. they do not originate from a single branch of the cell lineage tree). This can result from biological phenomena such as self-organization and context-dependent processes. Moreover, distantly related cells may converge to similar expression profiles due to functional similarity. For instance, in *C. elegans*, transcriptomic similarity initially correlates with lineage but diverges after gastrulation, illustrating how lineage signals can be lost as cells commit to specialized fates (Packer et al., 2019; Qiu et al., 2022; Bandler et al., 2022).

Despite these complications, transcriptomic data has been explored as auxiliary information for genetic lineage reconstruction (Zafar et al., 2020; Pan et al., 2023). Certain gene

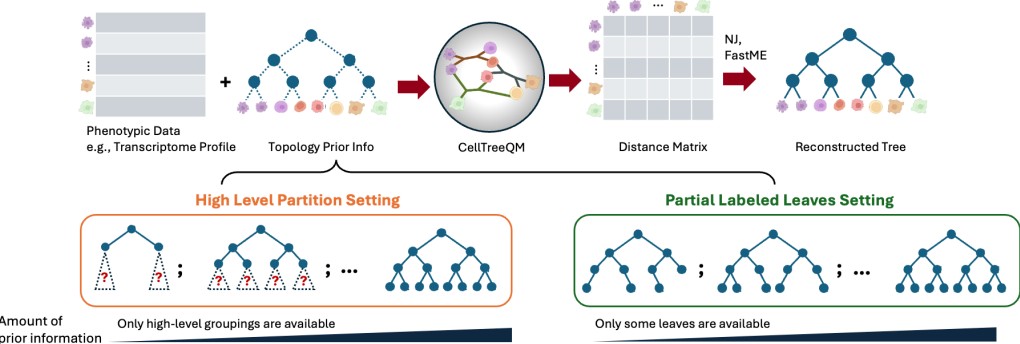

*Figure 2.* Overview of CellTreeQM Workflow for Lineage Reconstruction Using Metric Learning

expression patterns have been successfully leveraged to reconstruct portions of multicellular structures (Phillips et al., 2019; Shaffer et al., 2020; Rückert et al., 2022; Mold et al., 2024).

While the overall phenotype of the cell may not be consistent with cell lineage history, since cell lineages make history-dependent key decisions during developmental differentiation, the cell is likely to contain some kind of history information within its complex molecular states.

**Limited ground-truth lineage annotations** A second major challenge is the lack of comprehensive labeled datasets for training and evaluating lineage-reconstruction methods. Although technologies like CRISPR-based barcoding enable cell lineage labeling in certain contexts, they remain impractical or ethically restricted for many organisms (McKenna & Gagnon, 2019; Zafar et al., 2020). Consequently, only a handful of single-cell datasets include comprehensive lineage annotations. *C. elegans* is a rare exception: through painstaking observation and direct tracking of cell divisions in the 1970s and 1980s, Brenner, Sulston, and colleagues established its complete embryonic lineage (Sulston et al., 1983). Beyond this worm model, fully resolved lineage annotations are virtually nonexistent in other species, and even partial annotations—where only certain branches or cell types are labeled—are both scarce and often incomplete (Domcke & Shendure, 2023).

A central reason for this scarcity is the inherent difficulty of reliably assigning lineage labels in single-cell data. There are multiple biological barriers. First, the majority of cell division and differentiation happens inside the body where, except in some special cases, observation without sacrificing the animal is impossible. Second, cells are typically around $10^{-5}$ meters (10 micrometers) in size, making both visible traits and molecular measurements technically challenging and prone to noise. Lastly, the number of cells in a typical multicellular organism is extremely large—for example, the human body comprises approximately $10^{14}$ cells—making tracking individual lineages extremely difficult.

## 3. Related Works

**Metric Learning** Metric learning, also known as contrastive learning in the field of deep learning, has gained significant attention in recent years. These methods have been applied in both supervised and unsupervised settings, with the overarching goal of learning an embedding space where the distance between embedded points corresponds to a desired similarity measure or encodes semantic meaning. In this paper, we compare two widely used metric learning losses—triplet loss (Schroff et al., 2015) and quadruplet loss (Chen et al., 2017)—against our proposed approach.

In the context of single-cell biology, contrastive learning methods have been applied to scRNA-seq data to construct embeddings that capture nuanced cellular states (Yang et al., 2022; Heimberg et al., 2024). However, most existing approaches focus on cell-type classification or batch-effect correction rather than hierarchical structure or lineage reconstruction, leaving an open question on how contrastive learning can be effectively leveraged for lineage inference.

Lastly, theoretical results (Bartal, 1996; 1998) have provided constructions that bound graph metrics by expectations over distributions of tree graphs. Observed empirical metrics or metrics in learned latent space could be seen as implying a general graph metric and thus being associated with a distribution over tree graphs. Such a distribution might be seen as a probabilistic estimate of a cell lineage tree.

**Computational Methods for Cell Lineage** Phylogenetic inference seeks to reconstruct evolutionary relationships among species or cells. The parameter space is vast and inherently complex due to the combination of discrete topologies and continuous branch lengths, making the problem NP-hard in most formulations (Felsenstein, 2003). Various reconstruction algorithms employing different heuristics and data assumptions have been developed and have achieved reasonable performance on typical phylogenetic datasets in practice. The most widely used approach relies on heritable barcodes (e.g., CRISPR-based recorders) and infers

trees using maximum parsimony or distance-based methods (Jones et al., 2020; Gong et al., 2022). Our method falls within the category of distance-based lineage reconstruction, where the objective is to fit the data to its closest tree metric. Such problems are known to be NP-hard for $\ell_1$, $\ell_2$, and $\ell_\infty$ metrics on unrooted trees. Nonetheless, substantial effort has been devoted to improving approximation algorithms for these problems (Ailon & Charikar, 2005).

Recently, several studies have explored integrating gene expression data with barcode-based lineage inference to enhance reconstruction accuracy (Zafar et al., 2020; Pan et al., 2023). Notably, the most recent work (Schlüter & Uhler, 2025) investigates whether gene expression alone contains sufficient information for lineage reconstruction and reports promising results. In our study, we conduct similar permutation experiments to assess the feasibility of our approach. To the best of our knowledge, we are the first to reconstruct lineage solely from gene expression data and to curate benchmarks for this task under diverse scenarios and varying degrees of supervision.

# 4. Problem Formulation: Cell Lineage Reconstruction

## 4.1. Distance-Based Lineage Reconstruction

Given a dissimilarity matrix $D$, the goal of a distance-based approach is to solve the following optimization problem:

$$\min_{T \in \mathcal{T}} \|D - D_T\|_2,$$

where $\mathcal{T}$ denotes the space of all tree metrics, also called additive distance matrices. This problem is known to be NP-hard in the general case. However, when the deviation from a perfect tree metric is small, efficient algorithms exist that reconstruct the exact optimal tree. For instance, a sufficient condition for the Neighbor-Joining (NJ) algorithm (Saitou & Nei, 1987) to reconstruct the optimal tree is:

$$\|d' - d_T\|_\infty \leq x^*/2,$$

where $x^*$ denotes the shortest edge length in $T$ (Atteson, 1999).

A motivation for this distance-based formulation arises from the fact that leaf data generated by a Markov tree stochastic process will have its $\ell_2$ distance close to an additive matrix. Formally, we assume a vector-valued Markov process on the tree graph, where for a vertex:

$$\text{prob}(x_i | \text{parent}(x_i), t_i),$$

$x_i$ is the random state variable, $t_i$ is the time scaling to the parent vertex for the $i$-th element. We assume this probability is well-defined, either by a continuous-time finite-state Markov process or by a continuous-time Brownian motion process. Under this assumption, the expected squared Eu-

clidean distance matrix of the vertices is additive (Chang, 1996).

**Lemma 4.1.** *(Additivity of Expected Distances) For data generated by a Markov process on a tree $T$, the expected squared Euclidean distance between two leaves $i$ and $j$ satisfies the additive distance property:*

$$\mathbb{E}\big[\|\mathbf{x}_i - \mathbf{x}_j\|^2\big] = D_T(i,j),$$

*where $D_T(i,j)$ is the shortest path distance in $T$.*

In practice, real phenotypic data often violate these assumptions due to non-heritable effects, measurement noise, and convergent gene expression. These factors can be more problematic than the challenge of fitting an optimal tree, yet they have received limited attention in the literature. This is likely because phylogenetic data are typically preprocessed and curated by experts, whereas phenotypic data lack such standardized cleaning.

Thus, instead of relying solely on tree-fitting methods, we propose learning an embedding function that maps data points to a space where distances approximate an additive metric. Formally, given phenotype vectors $\boldsymbol{x}_i \in \mathbb{R}^p$ for leaf vertices $i$, we jointly search for a tree topology $T$ and an embedding function $f : \mathbb{R}^p \to \mathbb{R}^d$ that optimizes:

$$\min_{f,T} \|D(f(x)) - D_T\|_2^2 + \lambda \Omega(f),$$

where $\Omega(f)$ is a regularization term that prevents overfitting, weighted by $\lambda$.

## 4.2. Scenarios

**Supervised setting.** We assume that we have the *true* additive distances of the edge-weighted tree for every pair of leaves. The model is trained to embed the leaves such that $\|\mathbf{z}_i - \mathbf{z}_j\|$ is proportional to the known tree distances. At test time, the task is to recover the lineage relationships of new data using the learned embedding.

**Weakly Supervised Setting.** In many biological contexts, we do not have full access to the ground-truth lineage. Instead, only partial or coarser annotations are available, and we aim to generalize to the unknown vertices. We highlight two practical cases:

- *High-Level Partition Setting.* Biologists may know a small number of large "clades" (subtrees). For example, certain groups of cells are known to belong to distinct subtrees of the full tree.

- *Partial-Leaf Labeled Setting.* Some cells (vertices) have identified lineage tree graphs (e.g., mother–daughter cell pairs) obtained from, say, direct lineage-tracing experiments, but such information is absent for the majority of cells.

**Unsupervised Setting.** No explicit lineage information is

available at all. The model relies solely on the raw phenotypic data and data-driven estimation of desired metric properties of the embedding. This scenario is the most challenging because the algorithm must disentangle lineage-related signals from confounding variation in the data without any direct lineage cues.

### 4.3. Constructing the Tree

After learning the embeddings $\{z_i\}$ in which pairwise distances approximate the true tree distances, the next step is to build the lineage tree $T$. We use the NJ algorithm by default, which runs in time $O(n^3)$ for $n$ taxa. This is typically manageable for moderate dataset sizes. More discussion about phylogeny reconstruction is in §C. Additionally, we present an experimental comparison of different reconstruction methods under the supervised setting in §G.3.

### 4.4. Evaluation Metrics

We evaluate the reconstructed tree $\hat{T}$ against the ground-truth tree $T$ using the following metrics. Additional details can be found in §G.2.

**Robinson–Foulds (RF) Distance.** The Robinson–Foulds (RF) distance quantifies the topological difference between two unrooted trees by comparing their sets of partitions, where a partition corresponds to a bipartition of the taxa induced by an internal edge. The RF distance counts the number of partitions that differ between the inferred and true trees. We report the normalized RF distance, where 0 indicates identical tree topologies and 1 implies that no partitions are shared between the two trees.

**Quartet Distance (QD).** Although the RF distance is widely used in the literature, it is known to be insensitive to finer-grained subtree structures. To complement RF distance, we also report the Quartet Distance (QD) (Bryant et al., 2000). Given any four leaf vertices (i.e., a quartet of leaves), there are three possible unrooted tree topologies that describe their relationships. The quartet distance is computed as the fraction of quartets that are resolved differently in the two trees. For large trees, the quartet distance is approximated by randomly sampling a subset of leaf quartets.

## 5. Proposed Framework: *CellTreeQM*

We present *CellTreeQM* (Cell Tree Quartet Metric), a framework designed to reconstruct lineage relationships from phenotypic data. Our main objective is to learn an embedding function based on the assumption of additive pairwise distances. Our approach is to consider quartets of leaves and learn an embedding that optimizes tree-metric properties of known quartets in the latent space.

### 5.1. Additivity Loss via the Four-Point Condition

The additivity loss is derived from a classic property of an additive distance matrix known as the *four-point condition* (Theorem C.1). Denote a quartet of leaf vertices $i, j, k, l$ and define the following distance sums:

$$S_1 = D(z_i, z_j) + D(z_k, z_l),$$
$$S_2 = D(z_i, z_k) + D(z_j, z_l),$$
$$S_3 = D(z_i, z_l) + D(z_j, z_k).$$

If $D$ is an additive tree distance matrix, two of these sums match exactly and both exceed the remaining sum. In addition, the ordering of all quartets defines a unique unrooted tree topology. Considering the relaxed case where $D(\cdot, \cdot)$ are not additive, we define terms that measure deviation from additivity. Assume we have an ordering $S_1 \geq S_2 \geq S_3$. We define:

- $\mathcal{L}_{\text{close}}$ measures the gap between the top two sums and as a loss function encourages the two terms to be equal,

$$\mathcal{L}_{\text{close}} = |S_1 - S_2|.$$

- $\mathcal{L}_{\text{push}}$ enforces a margin so that the smallest sum ($S_3$) is sufficiently smaller than the average of the top two,

$$\mathcal{L}_{\text{push}} = \left[ S_3 - \frac{S_1 + S_2}{2} + m_0 \right]_+,$$

where $m_0 > 0$ is a margin hyperparameter.

We combine them into a single quartet loss: $\mathcal{L}_{\text{quartet}} = \mathcal{L}_{\text{close}} + \mathcal{L}_{\text{push}}$, then average over all (or a sampled subset of) quartets $Q$ in each training batch:

$$\mathcal{L}_{\text{additivity}} = \frac{1}{Q} \sum_Q \mathcal{L}_{\text{quartet}}$$

Figure 3 illustrates the geometric intuition of the loss. Under ideal additivity, the four points form an unrooted tree (Figure 3a), where $S_1 = D(A, C) + D(B, D)$ and $S_2 = D(A, D) + D(B, C)$ and $S_1 = S_2$. When additivity is violated, this balance is disrupted, and the structure can be imagined as a "box" with an extra edge (Figure 3b). The $\mathcal{L}_{\text{close}}$ term encourages the top two distance sums to become more similar, thereby reducing the asymmetry that creates the box-like distortion. In effect, it ensures that the box is not "fat". Meanwhile, the $\mathcal{L}_{\text{push}}$ term increases the gap between the smallest sum and the average of the top two, effectively "widening the bridge." This widening enhances the tree model's robustness to noise and distortions. We note that the ordering $S_1 \geq S_2 \geq S_3$ can be derived from supervised knowledge of the true quartet tree or from computations of the empirical data in the latent space.

### 5.2. Regularization: Deviation Loss

To prevent the learned embedding from drifting too far from the original data, we introduce a deviation loss $\Omega$. This loss

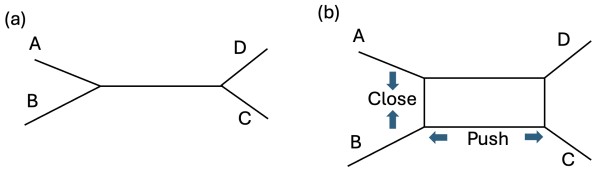

*Figure 3.* Geometric Intuition of the Quartet Loss

penalizes large discrepancies between the original and the induced distance matrix, $D(X)$ and $D(f(X))$, respectively, in the latent space:

$$\Omega(f, X) = \frac{1}{N}\|\mathcal{D}(f(X)) - \mathcal{D}(X)\|_F^2$$

where $\|\cdot\|_F^2$ denotes the squared Frobenius norm and $N$ is the number of points. This ensures that the latent space remains faithful to the measured phenotypic similarities and also prevents the learning process from degenerating in scale.

### 5.3. Feature Gating

Many high-dimensional phenotypic datasets (e.g., scRNA-seq) include numerous features that do not reflect lineage-related variation. To address this, we introduce a *feature gating* module, which adaptively modulates the contribution of each input feature. By emphasizing lineage-relevant signals and down-weighting confounding or redundant attributes, the gating module can improve downstream tree reconstruction. To learn an effective feature mask, we use a Gumbel-Softmax (Gumbel, 1954) approach that promotes discrete gating decisions (see §F.2). The gating is applied via a simple elementwise product: $\tilde{x}_i = x_i \cdot g_i$.

**Integrated Objective.** Combining Gumbel gating with our core metric-learning objectives, we arrive at the overall optimization function:

$$\min_{f,g}\left[\mathcal{L}_{\text{additivity}}(f \circ g, X) + \lambda\,\Omega(f \circ g, X) + \Omega_{\text{gates}}^{(\text{sparsity})}\right].$$

This integrated framework allows us to *jointly* learn which features are most informative for lineage reconstruction and how to embed them to satisfy the quartet-based additivity constraints.

### 5.4. Model Architecture

Our proposed framework, *CellTreeQM*, aims to learn embeddings from high-dimensional phenotypic data that facilitate phylogenetic reconstruction. To effectively learn relationships among cells, we use a sequence of Transformer encoder blocks as the backbone of the network, illustrated in Figure 11. Unlike classical Transformer models, we do not include positional encodings, as our input leaves are not inherently ordered. Without positional constraints, the self-attention module focuses purely on learning meaning-

ful relationships based on the feature similarities between cells. We compared the performance of networks with Transformer encoders as backbones versus those using fully connected layers, under a supervised setting on two *C. elegans* real datasets (see Table.5 in the Appendix). Networks with attention modules performed much better than the fully connected counterparts.

## 6. Cell Lineage Reconstruction Benchmark

To systematically evaluate lineage-reconstruction methods, we introduce a benchmark comprising a synthetic dataset and three scales of lineage-resolved real datasets. Figure 4 shows the training dynamics of CellTreeQM on both the synthetic and real datasets under the supervised setting. §D provides detailed dataset descriptions, parameter choices, and curation procedures.

**Lineage-Resolved *C. elegans* Dataset:** Among model organisms, *C. elegans* is uniquely suited for benchmarking lineage reconstruction because its embryonic cell lineage is *invariant*. We curate three subsets of increasing size–*C. elegans Small*, *Mid*, and *Large*–from transcriptomic atlases by Packer et al. (2019) and Large et al. (2024), containing 102, 183, and 295 leaves, respectively. Each subset is fully lineage-resolved, providing ground-truth tree topologies. These datasets allow us to measure reconstruction quality at multiple scales.

**Synthetic Brownian-Motion Simulations:** We generate synthetic datasets by simulating random branching trees and evolving feature vectors along branches via Brownian motion. To better reflect real-world noise, we add independent Gaussian noise and "alternative-tree" features that follow a separate confounding lineage. By tuning parameters such as the number of leaves and the relative strengths of signal and noise, we generate datasets with varying levels of reconstruction difficulty.

## 7. Experiments

### 7.1. Baselines

We consider the following two common contrastive losses in metric learning. The details can be found in §G.1. *CellTreeQM* and baselines produce learned embedding spaces from which we extract pairwise distances between leaves.

**Triplet Loss:** We designate the closest pair from each quartet as anchor–positive and the farthest leaf as negative. The loss encourages $\|f(A) - f(P)\| < \|f(A) - f(N)\|$.

**Quadruplet Loss:** Extends Triplet Loss by introducing a second negative leaf, enforcing additional pairwise margins to improve global distance structure.

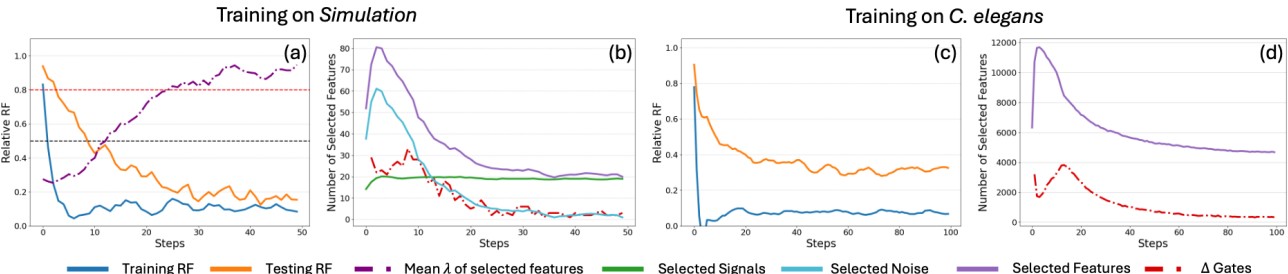

*Figure 4.* Supervised training dynamics on simulation and *C. elegans* Small dataset. Dashed purple is the Pagel's $\lambda$ for Phylogeny signal.

*Table 1.* **Supervised results on the *C. elegans Small* dataset.** Direct reconstruction on raw data yields RF = 0.923 and QDist = 0.554. Suffix "-G" denotes feature gating, and "-p" indicates label permutation. The reported values are means across three runs, with standard deviations in parentheses.

|  | Train RF $\downarrow$ | $\Delta\%$RF $\uparrow$ | $\Delta\%$QDist $\uparrow$ |
|---|---|---|---|
| CellTreeQM | **0.000** (0.00) | **0.690** (0.05) | **0.867** (0.02) |
| CellTreeQM-G | **0.000** (0.00) | **0.757** (0.03) | **0.848** (0.01) |
| CellTreeQM-p | 0.013 (0.00) | 0.434 (0.04) | 0.691 (0.05) |
| Triplet | 0.519 (0.03) | 0.179 (0.01) | 0.637 (0.02) |
| Triplet-G | 0.545 (0.03) | 0.203 (0.03) | 0.631 (0.02) |
| Triplet-p | 0.677 (0.06) | 0.037 (0.01) | 0.385 (0.06) |
| Quadruplet | 0.057 (0.00) | 0.454 (0.02) | 0.784 (0.01) |
| Quadruplet-G | 0.061 (0.00) | 0.484 (0.04) | 0.791 (0.01) |
| Quadruplet-p | 0.118 (0.00) | 0.149 (0.01) | 0.538 (0.02) |

## 7.2. Supervised setting

Similar to (Schlüter & Uhler, 2025), we conduct experiments in a supervised scenario where all quartet topologies are assumed known and we train a model that enforces such topology in the embedding space. While this setting has little empirical significance, its results justify the soundness of our approach. We use $\Delta\%$RF and $\Delta\%$QD (see Eq.3) to measure relative improvement over raw data and Training RF to assess how well the learned embedding preserves the global tree structure implied by the quartets.

Table 1 shows results on the *C. elegans Small* dataset, whose direct reconstruction yields high RF (0.923), suggesting almost no lineage information. We train each method both with and without feature gating (*-G*). Additionally, to test whether our method can overfit to any given topology, we randomly permute (*-p*) the labels of the leaves of $T$ to obtain an arbitrary tree $T'$ for training. Several key trends emerge:

**CellTreeQM consistently outperforms contrastive losses.** By explicitly enforcing the four-point condition across quartets, *CellTreeQM* consistently achieves higher improvement over raw data ($\Delta\%$RF and $\Delta\%$QD) compared to the triplet and quadruplet baselines. Moreover, triplet loss can recover a moderate level of quartet topology, but its latent space is not aligned on the global tree topology with large

Train RF and low $\Delta\%$RF. This is reasonable because neither contrastive loss explicitly enforces tree topology in the embedding space. Additionally, feature gating (*-G*) typically yields additional gains, especially in RF distance, likely by pruning out non-heritable or noisy gene expressions.

**Permutation experiments validate quartet-based fitting.** Training *CellTreeQM* on the random tree $T'$ still embeds the data with moderate fidelity, but less accurately than with the true tree. In contrast, triplet and quadruplet models struggle to align a random tree with the data in a similar manner. Similar trends hold for the *C. elegans* Mid and Large datasets, where triplet/quadruplet losses improve over raw data but still lag behind *CellTreeQM* (see Table 8). These results echo the recent findings in (Schlüter & Uhler, 2025) showing that *true lineage topologies* yield lower training losses and more generalizable embeddings, thus confirming there is ample lineage signal in the gene expression data—our method is not merely overfitting.

Figure 12 (2D t-SNE) shows that *CellTreeQM* better preserves the hierarchical structure than Triplet or Quadruplet. Figure 13 further illustrates the reconstructed trees, where *CellTreeQM* yields lineage-consistent subtrees.

## 7.3. Weakly Supervised

**High-Level partition setting.** A *level* is defined as the number of branching steps from the root. We assume we know how leaves are divided into clades at each level, as well as the relationships among these clades (see Fig. 2 left). Hence, each leaf is assigned to exactly one of these high-level clades, but the tree structure *within* for each clade remains unknown. Suppose we know that the leaves partition into four groups. Given a quartet, if there is at least one pair of leaves that exactly belong to one group, we can uniquely determine the correct topology among the three possible unrooted trees. We call these "known quartets". Otherwise, we call it an "unknown" quartet.

We compute the additivity loss *only* on the known quartets. We then evaluate on the full set of leaves, measuring the overall RF distance and quartet distance (QD) against the

*Table 2.* **Weakly supervised results on *C. elegans Small* under different partition levels.** K-QD and U-QD are quartet distances on the *known* and *unknown* quartets, respectively. The reported values are means across five runs, with standard deviations in parentheses.

| Method | $\Delta\%$RF↑ | $\Delta\%$QD↑ | $\Delta\%$K-QD↑ | $\Delta\%$U-QD↑ |
|---|---|---|---|---|
| **Partition Level: 3** | | | | |
| CellTreeQM | **0.349** (0.02) | **0.849** (0.01) | **0.994** (0.00) | 0.182 (0.03) |
| Triplet | 0.193 (0.02) | 0.521 (0.01) | 0.582 (0.02) | **0.241** (0.00) |
| Quadruplet | 0.090 (0.02) | 0.619 (0.01) | 0.768 (0.01) | -0.065 (0.01) |
| **Partition Level: 2** | | | | |
| CellTreeQM | **0.274** (0.03) | **0.805** (0.02) | **0.998** (0.00) | **0.558** (0.04) |
| Triplet | 0.191 (0.01) | 0.399 (0.01) | 0.623 (0.01) | 0.111 (0.01) |
| Quadruplet | 0.058 (0.01) | 0.485 (0.01) | 0.894 (0.01) | -0.037 (0.01) |
| **Partition Level: 1** | | | | |
| CellTreeQM | **0.164** (0.01) | **0.631** (0.05) | **0.997** (0.00) | **0.485** (0.07) |
| Triplet | 0.120 (0.02) | 0.356 (0.01) | 0.832 (0.01) | 0.167 (0.02) |
| Quadruplet | 0.070 (0.02) | 0.227 (0.01) | 0.678 (0.02) | 0.046 (0.01) |

ground-truth tree. Additionally, we separate quartets into known versus unknown and track QD for each group (K-QD for known quartets, U-QD for unknown).

Table 2 summarizes performance for varying levels on the *C. elegans Small* dataset. In all cases, *CellTreeQM* outperforms contrastive baselines: it more effectively leverages the known quartets while generalizing better to the unknown ones. By contrast, the quadruplet loss fails to generalize to unknown quartets and can even underperform the raw distance. At level 3, triplet loss does better on the unknown quartets than *CellTreeQM*, but it does so at the cost to the known quartets, illustrating its difficulty in balancing partial lineage supervision with global tree structure.

**Partial-Labeled Leaves Setting.** In this scenario, only a subset of leaves (e.g., 30%, 50%, or 80%) have known lineage information, while the remaining leaves are unlabeled (see Fig. 2 right). We classify quartets as *Known* when all four leaves are labeled, *Unknown* if none are labeled, and *Partial* otherwise. During training, we only apply the additivity loss to Known Quartets. After training, we embed *all* leaves and compute overall RF distance and QD, alongside three specialized QD metrics for Known (K-QD), Partial (P-QD), and Unknown (U-QD) quartets.

As shown in Table 3, *CellTreeQM* outperforms the contrastive baselines across all labeling fractions, particularly at 80% and 50%. At 30% labeling, however, all methods yield limited improvement on P-QD and U-QD, highlighting the difficulty of reconstructing lineage from randomly sampled sparse supervision. Notably, although the *Quadruplet* loss tended to overfit in the high-level partition scenario, it performs strongly here, surpassing *Triplet* and approaching *CellTreeQM* at moderate labeling fractions.

**Additional results.** To further validate *CellTreeQM*, we evaluated the model on two additional CRISPR-based mESC lineage datasets within the high-level partition setting. Results are provided in §I.

### 7.4. Unsupervised Setting

Lastly, we present preliminary results under the unsupervised regime, where no quartet constraints or partial subtree information are given. Here, we use a data-driven estimate of quartet order in the latent space. *CellTreeQM* can still learn a representation that partially adheres to tree-metric properties in the constrained setting of the simulation data, but the performance on real data was limited, suggesting better strategies are needed. See details in §G.5.

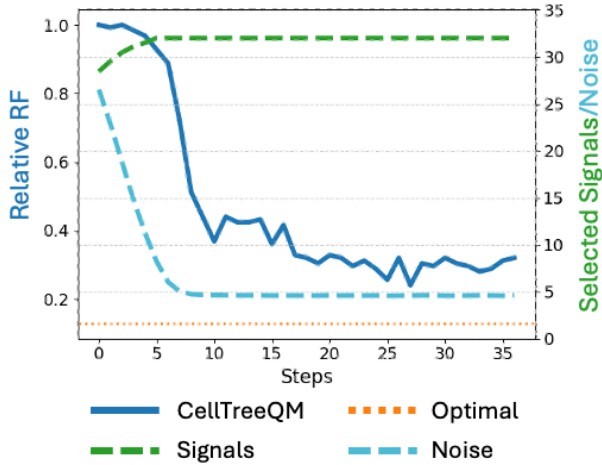

*Figure 5.* **Training dynamics of CellTreeQM in a purely unsupervised setting on a simulated dataset.** Optimal is the RF of reconstructed tree only based on signal features.

## 8. Discussion

**Summary.** We introduce *CellTreeQM*, a deep learning framework for reconstructing cell lineage trees from phenotypic features via metric learning. The key idea is that certain geometric structures are well-suited for representing tree-graph relationships. Therefore, we formulate our problem as learning a latent space whose metric properties are optimized for quartets of tree-graphs. To our knowledge, this is the first method to cast cell lineage inference explicitly as a metric learning problem. Empirical results in supervised and weakly supervised settings show that *CellTreeQM* considerably improves lineage reconstruction accuracy over standard contrastive baselines (e.g., triplet and quadruplet losses). This suggests that transcriptomic data contain cryptic lineage information that can be uncovered with carefully designed metric learning models.

**Limitations.** Despite its advantages, *CellTreeQM* has several limitations. First, our loss function integrates tree-metric constraints, gating, and distortion regularization. Although the distortion term limits the deviation of the latent space geometry from input feature space, it can impede

*Table 3.* **Weakly supervised results on *C. elegans Small* across different known fractions.** K-QD, P-QD and U-QD are quartet distances on the *known*, *partial* and *unknown* quartets, respectively. The reported values are means across ten runs, with standard deviations in parentheses.

| Method | Train RF↓ | Δ%RF↑ | Δ%QD↑ | Δ%K-QD↑ | Δ%P-QD↑ | Δ%U-QD↑ |
|---|---|---|---|---|---|---|
| **Known Fraction: 0.8** | | | | | | |
| CellTreeQM | **0.024** (0.03) | **0.448** (0.06) | **0.842** (0.05) | **0.999** (0.00) | **0.742** (0.07) | **0.465** (0.13) |
| Triplet | 0.454 (0.06) | 0.175 (0.05) | 0.728 (0.03) | 0.895 (0.01) | 0.624 (0.05) | 0.339 (0.14) |
| Quadruplet | 0.066 (0.04) | 0.403 (0.03) | 0.796 (0.06) | 0.953 (0.02) | 0.697 (0.09) | 0.435 (0.22) |
| **Known Fraction: 0.5** | | | | | | |
| CellTreeQM | **0.012** (0.01) | 0.092 (0.05) | **0.609** (0.05) | **0.999** (0.00) | **0.598** (0.06) | **0.398** (0.05) |
| Triplet | 0.303 (0.09) | 0.049 (0.04) | 0.505 (0.05) | 0.879 (0.02) | 0.493 (0.05) | 0.304 (0.06) |
| Quadruplet | 0.023 (0.02) | 0.115 (0.04) | 0.549 (0.04) | 0.934 (0.03) | 0.537 (0.04) | 0.340 (0.06) |
| **Known Fraction: 0.3** | | | | | | |
| CellTreeQM | **0.000** (0.00) | -0.023 (0.02) | **0.368** (0.06) | **1.000** (0.00) | **0.401** (0.06) | 0.250 (0.06) |
| Triplet | 0.156 (0.06) | -0.001 (0.02) | 0.358 (0.04) | 0.889 (0.02) | 0.384 (0.04) | **0.263** (0.05) |
| Quadruplet | 0.008 (0.02) | -0.020 (0.03) | 0.358 (0.03) | 0.918 (0.04) | 0.386 (0.03) | 0.259 (0.02) |

learning when lineage signals in real data are significantly distorted. While gating helps suppress random noise in simulations, its impact on real data is mild, suggesting that real "noise" features may be correlated with signal features. Second, we rely on NJ for final tree construction. Although NJ is widely adopted, it can fail under complex noise conditions. Future research could explore more robust tree-inference methods, such as Bayesian approaches or graph neural networks. Third, our benchmarks focus on lineage-resolved datasets with well-defined ground truth. Extending *Cell-TreeQM* to more heterogeneous single-cell datasets (e.g., developmental atlases with partial lineage annotations) will be crucial for broader applicability.

**Guidance for Future Work.** The modest success of unsupervised CellTreeQM on small, clean simulations opens the door to future improvements. Additional data-driven methods for determining the optimal latent space geometry, as well as more extensive hyperparameter tuning, may further enhance performance. In real single-cell transcriptome datasets, the phenotype–lineage correlation can be weak, particularly at later stages of development; thus, stronger regularization or heuristic constraints might be needed. Nevertheless, our initial findings show that unsupervised approaches can capture coarse lineage structure without explicit topological supervision. We believe that learning metric properties of the latent space, rather than directly trying to infer the tree-graph, can be an effective approach.

## Acknowledgements

This work was partially supported by the National Institutes of Health (NIH) under grant R01HD105819.

## Impact Statement

This paper presents work whose goal is to advance the field of Machine Learning as applied to a specific biomedical problem. There might be potential societal consequences of our work related to enhancing the understanding of human diseases. But we do not foresee any potential impact that requires specific highlighting.

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

# A. Related Works

Beyond the core comparisons in the main text, we here provide a broader review of related works.

**Metric Learning**    Metric learning is a broad field focused on learning representations in which distances capture meaningful similarity relationships. In recent years, a subset of metric learning techniques known as contrastive learning has gained significant traction in deep learning. These methods have been applied in both supervised (Khosla et al., 2021) and unsupervised (He et al., 2020) settings, with the main objective being to embed similar examples closer together while pushing dissimilar ones farther apart. In this paper, we compare two widely used metric learning objectives—Triplet loss (Schroff et al., 2015) and Quadruplet loss (Chen et al., 2017)—against our proposed approach.

**Representation Learning for scRNA-seq**    ScRNA-seq data are high-dimensional and subject to diverse technical and biological noise. Common strategies to handle this complexity often involve dimensionality reduction (e.g., PCA (Heumos et al., 2023)), or deep generative models (e.g., scVI (Lopez et al., 2018)), which embed cells in a lower-dimensional space that preserves essential variability.

Recently, large-scale pretrained models have emerged in single-cell analysis. Transformer-based architectures such as Geneformer (Theodoris et al., 2023) and scGPT (Cui et al., 2024) learn embeddings from massive single-cell corpora, facilitating tasks such as cell-type classification, data integration, and cross-modality predictions. Meanwhile, supervised contrastive learning on scRNA-seq has been applied to capture nuanced cellular states, offering efficient data usage and strong generalization (Yang et al., 2022; Heimberg et al., 2024; Zhao et al., 2025).

Nevertheless, most existing approaches rely on a notion of similarity derived from overall transcriptomic profiles, which does not necessarily align with lineage relationships (as discussed in the next paragraph). Although these methods successfully map cellular states and correct for technical artifacts, they do not directly address how to exploit these representations for cell lineage reconstruction.

**Trajectory Inference vs. Lineage Reconstruction**    Methods such as RNA velocity (La Manno et al., 2018; Bergen et al., 2020) and trajectory inference (Qiu et al., 2017; Street et al., 2018) reveal continuous trajectories in the molecular state space of the transcriptomes. Although these tools capture *average* progression trends, they do not directly yield a cell-by-cell lineage hierarchy. Rather, they provide trajectory embeddings that broadly reflect state set evolution, rather than the lineage history of cells. Moreover, genes driving these gene trajectory embeddings are typically selected based on high global variance, potentially missing the key drivers for lineage-specific processes.

**Cell Types vs. Cell Linages**    In standard single-cell analysis, cell types are typically inferred by grouping cells with similar transcriptomic profiles, either through unsupervised clustering (Heumos et al., 2023) or reference-based classification methods (Aran et al., 2019; Ianevski et al., 2022; Hu et al., 2023). While these approaches effectively capture functional similarities, they do not explicitly account for the developmental origins of each cell. In other words, two cells with nearly identical gene expression patterns may not necessarily share a recent common ancestor. In contrast, cell lineages focus on the actual historical branching process by which cells emerge and differentiate—thus requiring methods that go beyond mere transcriptomic similarity to capture genealogical relationships.

**Phylogenetic Inference**    Reconstructing the lineage history of species or cells is a phylogenetic inference problem. The parameter space is vast and inherently complex due to the combination of discrete topologies and continuous branch lengths, making the problem NP-hard in most formulations (Felsenstein, 2003). Despite this, various data assumptions and heuristic-based reconstruction algorithms have been developed, achieving reasonable performance on typical phylogenetic datasets. The most widely used approaches infer trees using maximum parsimony, maximum likelihood, Bayesian inference, or distance-based methods (Jones et al., 2020; Gong et al., 2022).

Our method falls within the category of distance-based lineage reconstruction, where the goal is to fit the data to its closest tree metric. Finding the optimal tree metric is NP-hard under $\ell_1$, $\ell_2$, and $\ell_\infty$ norms for unrooted trees, but significant progress has been made in improving approximation algorithms (Ailon & Charikar, 2005). For instance, (De Soete, 1983) proposed a greedy approach by directly using gradient descent to find the closest additive distance matrix. However, instead of explicitly optimizing pairwise distances, our approach learns an embedding function that maps data points into a space where the geometric properties facilitate tree inference.

Notably, the most recent work (Schlüter & Uhler, 2025) investigates how to identify phylogenetically informative features under a fully supervised setting using permutation approaches. In our study, we conduct similar permutation experiments to assess the feasibility of our approach, but with the added objective of not only characterizing the features but also directly reconstructing the tree.

## B. Terminology

Tree graph algorithms in biology originate from the mathematical systematics field dealing with lineage reconstruction of whole organisms, represented by species or taxons, while the cell lineage reconstruction problem arises from the field of molecular and developmental biology. These two different areas have distinct but also overlapping terminology (Kim & Warnow, 1999; Clevers et al., 2017; Zeng, 2022; Domcke & Shendure, 2023; Rafelski & Theriot, 2024). Here, we establish the following definitions to avoid confusion.

**Single-Cell Biology**

- Cell state: Characterization of a cell's molecular phenotype. This phenotype typically varies with time and space, comprised of measurements of gene expression, metabolism, and other functional properties.

- Cell lineage: The sequential path of cell divisions that traces a given cell's ancestry back to the zygote. Some times, cell lineage is used to refer to parts of the path. For example, "neuronal lineage" might refer to the parts of the path that disticly lead to cell groups identified as neuronal cell type (see next).

- Cell type: A classification of cells based on cell phenotype and sometimes also cell lineages. In general, cells of the same type typically share functional and structural characteristics but does not necessarily imply lineage relationships. However, in some biological cases, a particular cell type might be established by its particular lineage relationship rather than just the cell phenotype.

**Phylogenetics and Evolutionary Biology**

- Phylogeny: A tree-structured graph describing the evolutionary history and relationships among a set of biological entities. In Systematics, these entities are typically taxons (see next). In particular, a phylogeny is typically a leaf-labeled tree graph, in the sense that only the leaves of the tree are measured and named entities and interior vertices are hypothetical unnamed ancestors.

    When the entities are cells, a phylogeny can be seen as a *cell lineage tree*.

- Cell lineage tree: A specialized tree-structured graph where each node represent a cell or a group of cells, and edges represent cell division events. The root corresponds to the common ancestral cell for all cells in the tree. Ultimately, all cells lead to the fertilized egg as the root. Branches typically depict temporal progression leading to changes in cell state termed differentiation. Sometimes, branches might represent a collection of cell divisions.

- Vertices: The nodes in a lineage tree. An *internal vertex* usually corresponds to a cell division point, producing two (or more) daughter cells. A *leaf vertex* (or *leaf*) typically represents a terminally differentiated cell. However, if the cells are sampled from middle of the cell replication process, the leaf vertex may represent transient cell(s).

- Taxon (plural: taxa): *taxon* is an abstract unit in Systematics, referring to a distinct group of organisms that has been annotated by an expert for the purposes of classification. A canonical example is a species or an isolated population. Here we use it to denote the biological entity represented by a vertex of the cell lineage tree. This entity might be a single cell or it might be a class of cells denoted as a cell type.

- Clade: A subset of a lineage tree that includes a common ancestor and all its descendant leaves.

## C. Preliminaries of Phylogeny

A tree is a connected graph where every pair of nodes is connected by a unique path. Among other things, this restriction implies that in a tree there are no links $(v_i, v_{i'})$ that connect a node $v_i$ to itself. An additive tree is a connected undirected

network where every pair of nodes is connected by a unique path(Sattath & Tversky, 1977). Since there exists only one path between any two nodes, the minimum path length distance between two nodes is equal to the length of the unique path that connects them. These distances are often referred to as path length distances or additive tree distances.

In evolutionary biology, a phylogeny is a bifurcating tree that models the evolutionary relationships among a set of species or other biological entities. The leaves of this tree represent extant (or observed) entities, while the internal nodes represent their hypothetical common ancestors. Such trees help us understand how these entities have diverged over time.

With the advent of large-scale genomic data, the field of phylogenomics has emerged. Phylogenomics integrates phylogenetic analysis with genome-wide data, allowing for more accurate and comprehensive inferences about evolutionary history. Rather than focusing on a single gene (as in classical phylogenetics), phylogenomics considers data from multiple genes, entire genomes, or high-dimensional molecular measurements, providing a richer context for reconstructing the evolutionary relationships among species.

In this section, we introduce fundamental concepts for distance-based phylogeny reconstruction—an approach that finds an additive tree by mapping $n \times n$ distance matrix to $n \times n$ additive distance matrix (Kim & Warnow, 1999; De Soete, 1983). These concepts form the theoretical basis of the learning objectives proposed in this work.

### C.1. Distance Matrices and Additivity

Given a set of $n$ nodes, each represented by a vector $\boldsymbol{x}_v$, we can derive an $n \times n$ distance matrix $M$, where $M_{ij}$ denotes the distance between nodes $i$ and $j$. If $M$ is a valid metric, it must satisfy the following properties:

- **Symmetric**: $M_{ij} = M_{ji}$ and $M_{ii} = 0$;

- **Triangle Inequality**: $M_{ij} + M_{jk} \geq M_{ik}$

Now, suppose $\boldsymbol{x}_v$ represents the leaves of a phylogeny and $w$ is the lowest common ancestor of two nodes $u$ and $v$, we expect $M_{uv} = M_{uw} + M_{wv}$. This property introduces the concept of an additive metric. A distance matrix $M$ is **additive** if there exists a phylogeny $T$ such that:

- Each edge $(u, v)$ in $T$ is associated with a positive edge weight $\delta_{uv}$.

- For every pair of nodes $u, v$, the distance $M_{uv}$ equals the sum of the edge weights along the unique path from $u$ to $v$ in $T$.

For distance matrices with fewer than four leaves, $M$ must be additive. However, for matrices with four or more leaves, $M$ may not be additive. Buneman's 4-point condition provides a criterion to determine additivity:

**Theorem C.1** (4-Point Condition). *A distance matrix $M$ is additive if and only if the following holds: For any distinct leaves $i, j, k, l$, we can label them such that:*

$$M_{ik} + M_{jl} = M_{il} + M_{jk} \geq M_{ij} + M_{kl}.$$

This condition ensures that, among any four leaves, the two largest sums of pairwise distances are equal. This property is fundamental for the additivity, guaranteeing that $M$ corresponds to an additive tree. When $M$ satisfies this condition, the additive tree can be uniquely reconstructed in $O(n^2)$ time.

### C.2. Non-Additivity and Approximate Solutions

In practical phylogenomics, observed distance matrices $M$ often deviate from perfect additivity. In such cases, the objective becomes finding an additive matrix $M_T$ that corresponds to a tree $T$ that minimizes the sum of squared errors (SSQ), defined as:

$$\min_T \|M_T - M\|_2.$$

It is known that finding the optimal $T$ is $NP$-hard for various norms ($p = 1, 2, \infty$) (Farach et al., 1995).

However, given a fixed tree topology $T$, one can at least solve for the optimal edge lengths $E$ that best fit $M$. This subproblem can be formulated as a non-negative least squares (NNLS) problem:

$$\min_{E \geq 0} \|P_T E - M_{\text{vec}}\|_2,$$

where :

- $E$ is an $m$-dimensional vector of edge lengths ($m$ is the number of edges in $T$).

- $M_{\text{vec}}$ the vectorized form of $M$, containing $\binom{n}{2}$ pairwise distances.

- $P_T$ is a $\binom{n}{2} \times m$ path matrix encoding the tree's topology. Each row corresponds to a pair $(i,j)$ of leaves, and each column corresponds to an edge $e$ in $T$. The entry $[P_T]_{(i,j);e}$ is 1 if edge $e$ lies on the path between $i$ and $j$, and 0 otherwise.

This NNLS problem is convex and can be solved efficiently. The difficulty lies in choosing the optimal topology $T$. Because the space of possible tree topologies grows super-exponentially. Since enumerating and evaluating all possible trees is not practical for larger $n$, heuristic methods are employed to approximate the solution. One of the widely used heuristics is the Neighbor-Joining (NJ) method (Saitou & Nei, 1987). The NJ algorithm is efficient and makes no assumptions about the edge lengths. As shown in (Gascuel, 1997), NJ reconstructs the unique tree when given an additive distance matrix. Moreover, (Atteson, 1999) proved that if a distance matrix $M$ is nearly additive, there exists an additive distance matrix $D_T$ such that:

$$|M - D_T|_\infty < \mu(T)/2$$

where $\mu(T)$ is the minimum edge length in $T$. All distance matrices $M$ that satisfy this condition share the same tree topology, meaning $T$ is the unique tree corresponding to these distances. The NJ algorithm has an optimal reconstruction radius in the sense that: (a) given a nearly additive distance function it reconstructs the unique tree $T$ and (b) there can be more than one tree for which $|M - D_T| < \delta$ holds if $\delta \geq \mu(T)/2$.

### C.3. From Genomic Data to Latent Representations

In classical phylogenomics, evolutionary distances are estimated from genomic sequences using probabilistic models. However, in the context of single-cell data, the observed distance matrix often deviates substantially from additivity due to factors such as measurement noise, high dimensionality, and features unrelated to lineage. In this work, we propose learning a nonlinear mapping that projects high-dimensional observations into a latent space, and then computing distances in that space. The goal is for these learned distances to reflect the underlying lineage structure more accurately than distances directly computed from the original data. In essence, we aim to transform empirical dissimilarities into additive phylogenetic distances, thereby bridging the gap between observed data and their development histories.

### C.4. Constructing the Tree with NJ

After learning the embeddings $\{z_i\}$ in which pairwise distances approximate the true tree distances, the next step is to build the lineage tree $\mathcal{T}$. A standard choice is the Neighbor-Joining (NJ) algorithm (Saitou & Nei, 1987), a greedy approach that iteratively merges pairs of nodes or clusters based on a pairwise distance matrix. Neighbor-Joining does not assume equal branch lengths and can yield accurate topologies even when distances are only approximately additive.

The algorithm is guaranteed to recover the correct tree topology when the distances perfectly adhere to an additive tree metric, and it often performs well even when this assumption is not strictly met. In general, the method has been shown to work well with finite datasets, and it is one of the most widely used distance methods for tree graph inference.

While NJ is a polynomial-time algorithm, its time complexity is $O(n^3)$ for $n$ taxa. This is typically manageable for moderate dataset sizes, but it may become computationally expensive for very large trees (Atteson, 1999; Mihaescu et al., 2009).

## D. Dataset Details

### D.1. Lineage-Resolved *C. elegans* Dataset

Among model organisms, the nematode *C. elegans* is uniquely suited for benchmarking lineage-reconstruction methods because its embryonic development follows an *invariant* pattern. Using the transcriptomic atlas by Packer et al. (2019) and (Large et al., 2024), we define three datasets with supervised tree-graph ground truth of varying sizes:

- **C. elegans Large (295 leaves).** This dataset, drawn directly from (Packer et al., 2019), provides a broad coverage of terminal lineages, offering a moderately sized yet comprehensive benchmark.

- **C. elegans Small (102 leaves).** To create a simpler, more tractable dataset, we prune the original *C. elegans* lineage to include only nodes that have a clearly defined mapping to annotated cell types. This smaller tree is useful for rapid prototyping and evaluating basic performance. Details of curation can be seen in later of this subsection.

- **C. elegans Mid (183 leaves).** Building on the lineage-resolved atlas for both *C. elegans* and *C. briggsae* by Large et al. (2024), we curate a set of 183 leaves that can be consistently mapped to both species. We use just the *C. elegans* portion here, providing an intermediate-sized dataset that balances coverage and complexity.

**Preprocessing** Following standard filtering criteria, we retain about 13,000 genes per dataset by removing those with minimal counts (fewer than 10 UMIs) or zero variance across cells.

**Summary Statistics.** Table 4 summarizes key statistics for each *C. elegans* dataset, including the number of cells per leaf, total leaves, Colless index (Lieberman & Wiley, 2011), tree diameter, depth, Faith's Phylogenetic Diversity (PD)(Faith, 1992), and mean pairwise distance.

The Colless index is a measure of tree imbalance, where a smaller value indicates a more balanced (symmetric) tree. Indeed, the relatively low Colless indices suggest that the *C. elegans* lineage is quite symmetric, and our pruned subsets remain well-distributed across the leaves. Faith's PD is a measure of biodiversity, defined as the sum of branch lengths in a phylogenetic tree.

We provide two sets of tree metrics. In the *Based on Reference Tree* section, branch lengths are set to be 1s in the complete reference tree (containing 669 leaves), and these lengths are retrained and accumulated while pruning so that the distances among leaves in our curated datasets reflect the distances in the complete reference tree. In the *Based on Topology* section, we recompute statistics under the assumption that all branch lengths are equal to 1 in the curated dataset, allowing us to evaluate the inherent structure of each pruned tree independently of the original branch lengths.

*Table 4.* **Statistics of Datasets in the Cell Lineage Tree Reconstruction Benchmark.** Values in parentheses indicate standard deviations where applicable.

| Dataset | N Cells/Leaf | Leaves | Colless | Based on Reference Tree | | | | Based on Topology | | | |
|---|---|---|---|---|---|---|---|---|---|---|---|
| | | | | Diameter | Depth | Faith's PD | Mean PD | Diameter | Depth | Faith's PD | Mean PD |
| C. elegans Small | 107 | 102 | 0.047 | 21 | 7.92 (2.07) | 293 | 15.52 | 19 | 6.66 (2.01) | 202 | 12.27 |
| C. elegans Mid | 101 | 183 | 0.018 | 19 | 8.21 (1.81) | 424 | 14.23 | 18 | 8.01 (1.66) | 364 | 13.53 |
| C. elegans Large | 117 | 295 | 0.010 | 20 | 7.91 (1.69) | 735 | 15.84 | 20 | 7.62 (1.64) | 588 | 14.77 |

**The curation of C. elegans Small.** In Packer et al. (2019), three cell ontologies are available: (1) cell identities by barcode, (2) cell type annotations (from Packer *et al.*), and (3) Lineage Node Names on the full embryonic tree (from Sulston *et al.*). As illustrated in Figure 6, the relationship between Cell Identities and Cell Type Annotations is many-to-one, determined by manual annotation. Meanwhile, the relationship between Cell Type Annotations and Lineage Node Names is many-to-many.

On one hand, a single Cell Type Annotation may correspond to multiple lineage nodes due to symmetry (e.g., *Cx* could be either *Ca* or *Cp*). On the other hand, one lineage node can be associated with multiple Cell Type Annotations because each annotation is defined as a distribution of cell states, which may overlap.

To create a relatively small, clean dataset while preserving the overall lineage structure of *C. elegans*, we selected only those cell type annotations that map to a single lineage node. We then constructed a lineage tree using these "clean" lineage nodes. This effectively serves as pruning the annotation tree. After pruning, we randomly drop a few clades to make sure the final lineage tree is in binary structure. As shown in Table 4, the curated C.elegans Small dataset spans most major cell lines of C. elegans while relatively easier to reconstruct than the full lineage tree.

### D.2. Simulation: Brownian Motion on Lineage Tree

We developed a synthetic dataset that encompasses both heritable and non-heritable features to model cell lineage relationships in a structured, probabilistic manner. Each leaf $u$ in this dataset is assigned three types of features:

$$\underbrace{(S_{u,1}, \ldots, S_{u,n_{\text{signal}}})}_{\text{signal variables}}, \quad \underbrace{(N_{u,1}, \ldots, N_{u,n_{\text{noise}}})}_{\text{Gaussian noise}}, \quad \underbrace{(A_{u,1}, \ldots, A_{u,n_{\text{AltSig}}})}_{\text{alternative-tree noise}},$$

providing a comprehensive representation that includes both true lineage signals and potential confounding factors.

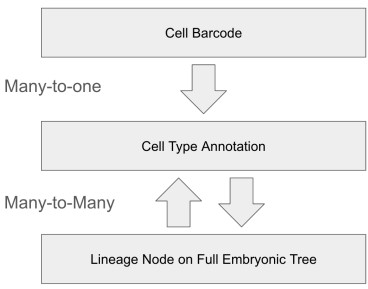

*Figure 6.* Entity relationships among the annotations in Packer et al. (2019).

### D.2.1. CELL LINEAGE SIGNALS

We represent the cell lineage as a full binary tree with $n_{\text{leaves}}$ terminal nodes. Each edge in the tree is assigned a length $t_j \sim \text{Uniform}(1, w_{\text{max}})$, which can be viewed as a developmental or temporal distance between parent and child nodes. To simulate heritable changes, we apply Brownian motion, a standard approach for continuous-character evolution (O'Meara, 2006; Pan et al., 2022).

Starting from a zero vector at the root, each child node's signal vector is obtained by adding a Gaussian increment $\mathcal{N}(0, t_j)$ to the parent's signal vector along the branch $j$. Let $i$ denote the index of one out of $n_{\text{signal}}$ Brownian motion realizations, thereby defining a signal feature $S_i$ that encodes lineage information. Formally, Brownian motion at edge $j$ is

$$S_i^{(\text{child})} = S_i^{(\text{parent})} + \mathcal{N}(0, t_j) \quad \text{for } i = 1, \dots, n_{\text{signal}}.$$

This part of simulation is governed by the hyperparameters $(n_{\text{signal}}, n_{\text{leaves}}, w_{\text{walk}})$. After generating all signal features, we compute the pairwise distance matrix and use Neighbor-Joining to reconstruct the lineage tree. The reconstructed tree is compared with the true tree using the Robinson-Foulds (RF) distance. A grid-sweep of the hyperparameters was conducted, and the results are presented in Fig. 8.

Figure 8 illustrates several key factors affecting lineage reconstruction accuracy. Increasing the number of signals $n_{\text{signals}}$ generally reduces the average RF distance, reflecting better reconstructions. Conversely, as the number of leaves $n_{\text{leaves}}$ grows, the RF distance tends to rise, indicating the challenges of reconstructing larger and more complex trees. The parameter $\log w_{\text{max}}$ can help distinguish lineages by amplifying their differences—particularly when $n_{\text{signals}}$ or $n_{\text{leaves}}$ is small—but its benefits come with a random variation. Consequently, beyond a certain threshold, increasing $\log w_{\text{max}}$ does not further reduce the RF distance; instead, an inherent "floor" of imperfect reconstruction persists. Finally, there is a saturation effect: beyond some point, neither adding more leaves nor more signals consistently pushes the RF distance toward zero, highlighting the limit imposed by Brownian motion's stochasticity and the maximum possible branch length $w_{\text{max}}$.

### D.2.2. INDEPENDENT GAUSSIAN NOISES

We introduce non-heritable noise features that are independent of the lineage. Each noise variable follows a Gaussian distribution with mean 0 and standard deviation $\sigma' = \beta \sigma_{\text{signal}}$. Symbolically,

$$N_i \sim \mathcal{N}(0, \sigma) \quad \text{for } i = 1, \dots, n_{\text{gaussian}}.$$

### D.2.3. ALTERNATIVE TREE NOISE

In addition to independent Gaussian noise, we introduce a second type of noise structured by an *alternative tree*. This *alternative tree noise*, or "anti-signal," represents confounding factors that give rise to correlated features unrelated to the true lineage. Examples include environmental or microenvironmental influences (e.g., regional nutrient availability), cell-cycle synchronization, or technical batch effects. These factors can cluster cells according to a topology distinct from the actual lineage, thus posing a challenge for tree reconstruction.

**Key parameters.** To generate alternative tree noise, we first partition the set of leaves into one or more subsets. In our running example, we split the leaves into two partitions of equal size, i.e., [0.5, 0.5]. For each partition, we construct

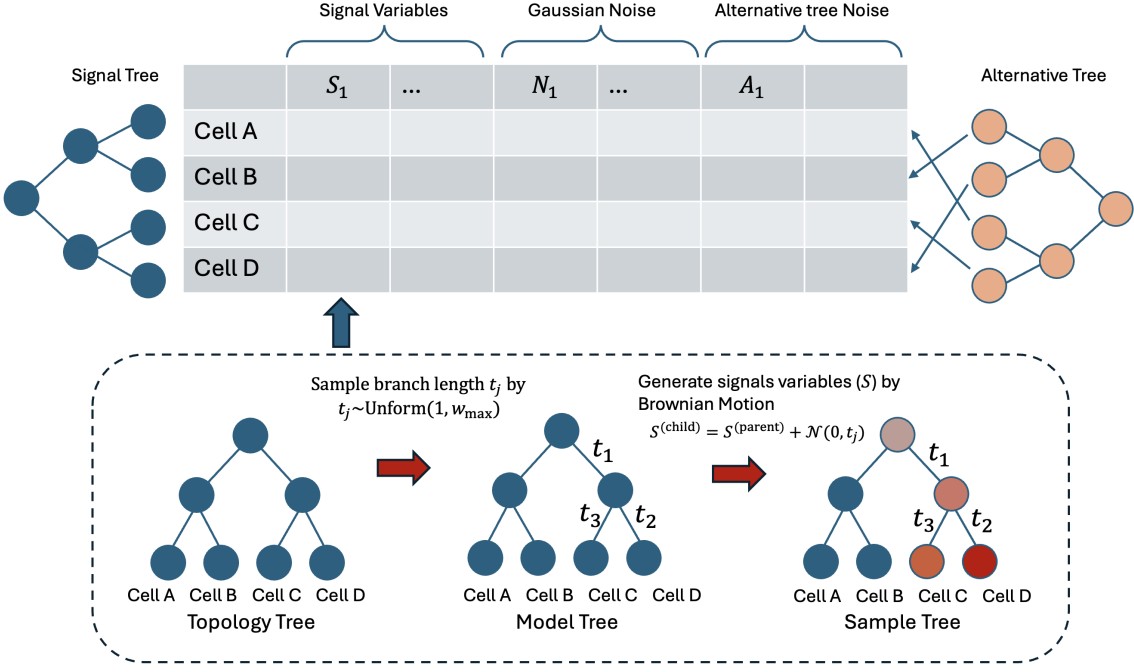

*Figure 7.* Overall schematic of the lineage tree simulation. The simulation starts with a *Topology Tree* defining cell relationships. Each edge is assigned a random length, producing a *Model Tree*. Signal variables ($S_i$) are then generated along the branches via Brownian motion process, resulting in the *Sample Tree*. Finally, Gaussian noise ($N_i$) and alternative tree noise ($A_i$) are added independently, yielding the complete feature matrix for each leaf. Alternative trees are constructed separately to simulate confounding topologies.

an alternative tree of a specified size (equal to the number of leaves in the largest partition, set to 4 in the example) and create a new set of "alternative" feature columns, AltSigs. Each leaf in the original (signal) tree thus has a corresponding "alternative" leaf in the alternative tree. We can optionally have multiple alternative trees per partition, but in the example, we set this number to 1.

**Generation procedure.** Similar to the main (signal) tree, each alternative tree has its own branch lengths drawn from some distribution (e.g., $\text{Uniform}(1, w_{\max}^{(\text{alt})})$). For every leaf u in the true (signal) tree, there is a corresponding leaf $u_l$ in the alternative tree. We then generate "alternative" features for each such leaf by simulating a Brownian-motion–like process, or another model as desired, along the branches of the alternative tree. These "anti-signals" are thus correlated according to the *alternative* topology rather than the true lineage topology. In the final dataset, each leaf u accumulates an additional vector of alternative noise features,

$$A_{u,1}, A_{u,2}, \ldots, A_{u,n_{\text{AltSig}}}.$$

Because these features reflect a different branching structure than the true cell lineage, they act as confounders in any tree reconstruction method that does not isolate genuine signals from extraneous structure. By controlling the number of partitions, leaves per alternative tree, number of alternative signals, and total number of alternative trees, one can tune the difficulty of distinguishing true lineage signal from these carefully structured "anti-signal" features. (Fig. 9)

### D.2.4. SUPERVISED SETTING

To evaluate lineage reconstruction methods in a supervised setting, we generate two sets of signal variables: one for training and one for testing (Fig. 10). First, we generate signal variables using the Brownian motion process described earlier. Then, for each signal variable, we add a small amount of Gaussian noise, $\mathcal{N}(0, 0.1 \times \bar{\sigma}_{\text{signal}})$, where $\bar{\sigma}_{\text{signal}}$ represents the mean standard deviation of the generated signal features across all leaves and signal dimensions. This creates a replicate of the signal variables with slight perturbations, suitable for evaluating a method's ability to generalize from training data to unseen test data. The training and testing datasets, therefore, have the same underlying lineage structure but differ in the specific

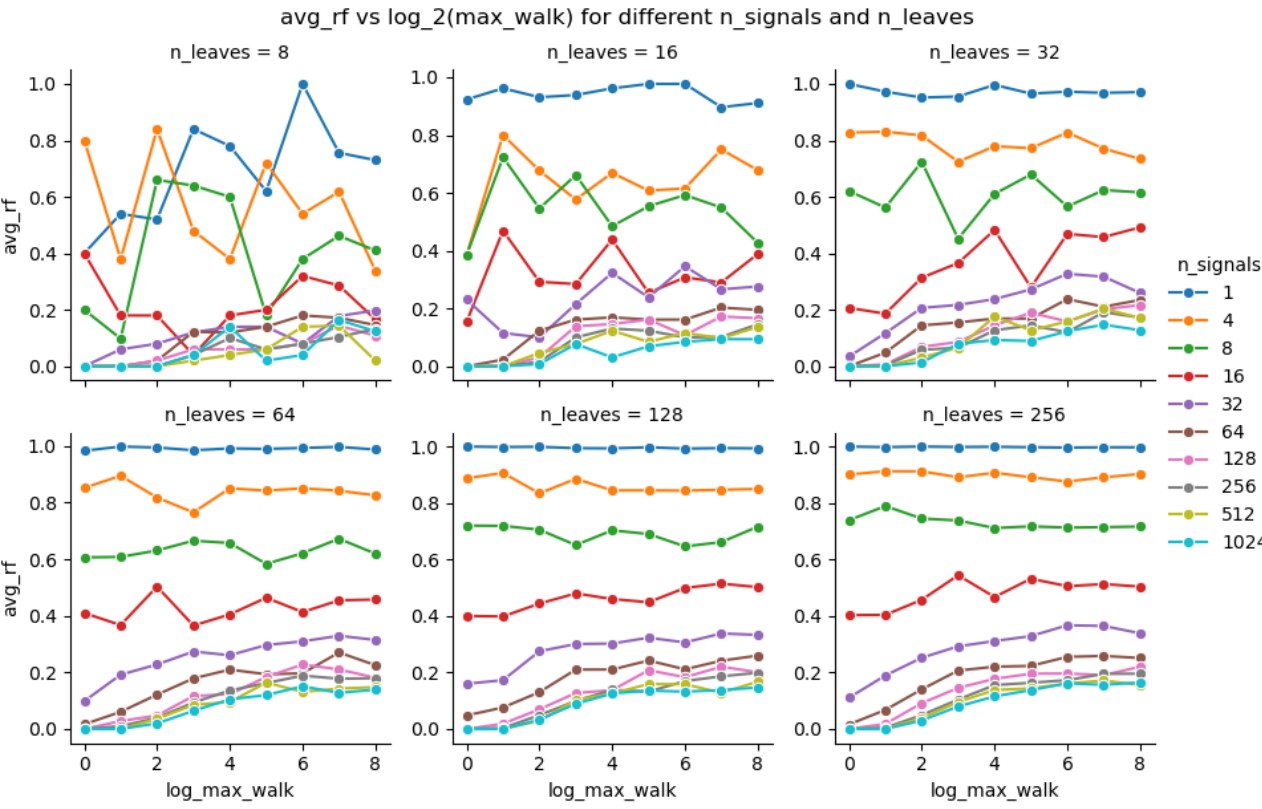

*Figure 8.* Mean Robinson-Foulds (RF) distance between reconstructed and true lineage trees as a function of $\log w_{\max}$ for varying numbers of signals $n_{\text{signal}}$ and leaves $n_{\text{leaves}}$. Each panel represents a different value of $n_{\text{leaves}}$, while distinct lines within each panel correspond to different numbers of signals. Here, $w_{\max}$ is the maximum possible edge length in the tree. Each data point shows the average RF distance across 25 trails (5 distinct model tree $\times$ 5 replicate sample tree per model tree.

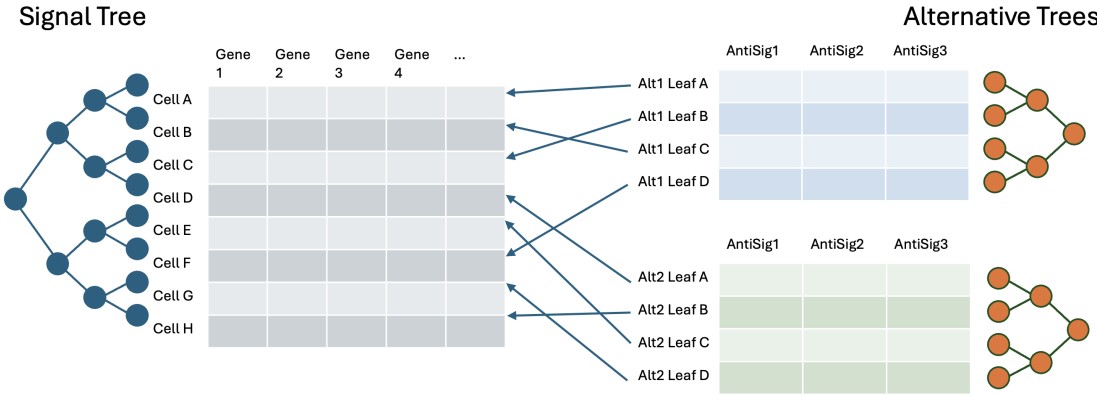

*Figure 9.* Schematic of Alternative Tree Noise Generation. The left panel depicts the true signal tree, representing the actual lineage relationships between cells. The middle panel shows the feature matrices for signal genes (left) and alternative tree noise or "anti-signals" (right). Arrows indicate the mapping of leaves from the signal tree to the corresponding leaves in the alternative trees. The right panel illustrates the alternative trees, which are constructed independently for each partition of leaves. These alternative trees have different topologies and branch lengths than the true signal tree, resulting in correlated noise features that can confound lineage reconstruction.

signal values due to the added noise. Gaussian and alternative tree noises are added independently to both the training and testing datasets as described in the previous subsections.

To complement real-data experiments, we generate synthetic datasets that capture both heritable and non-heritable features under controlled conditions (Fig.7):

- **Signal Features.** A "true" lineage (signal tree) is simulated with random branch lengths (edge-weights); We generated samples of Brownian motion along the edges where expected standard deviation of the stochastic process is parameterized by the branch lengths.

- **Independent Gaussian Noise.** Added to each leaf independently, these features are uncorrelated with the true lineage.

- **Alternative-Tree Noise.** Features are generated from a separate, "alternative" tree, modeling confounding patterns that could mislead reconstruction algorithms.

By tuning parameters (e.g., number of leaves, signal dimensionality, noise variance), we can systematically adjust difficulty and isolate key performance factors. Full details on the synthetic data generation, parameter choices, and hyperparameter sweeps appear §D.2 and Table 7.

## E. Problem Formulation Details

We consider an underlying lineage tree $\mathcal{T}$ that describes the data-generating process, where a single progenitor cell initiates binary replication while also modifying molecular states such that a diverse set of daughter cells is generated. In this bifurcating tree, each internal node corresponds to ancestral mother cells and edges represent mother-daughter relationships. The leaves of the tree correspond to sampled cells, typically terminally differentiated cells (*i.e.*, cells that no longer replicate). Our goal is to recover this tree structure $\mathcal{T}$ using high-dimensional phenotypic data, such as transcriptomic profiles, measured on the leaves.

### E.1. Prior Information

The problem of lineage reconstruction from single-cell phenotypic data poses challenges reminiscent of classical systematics in biology, a field historically concerned with organizing biological diversity based on observable traits. (Zeng, 2022; Domcke & Shendure, 2023; Quake, 2024) The low correlation between phenotype and cell lineage complicates the task. Selecting signal features is NP-hard; reconstructing the phylogeny is also a well-known NP-hard problem. But we are

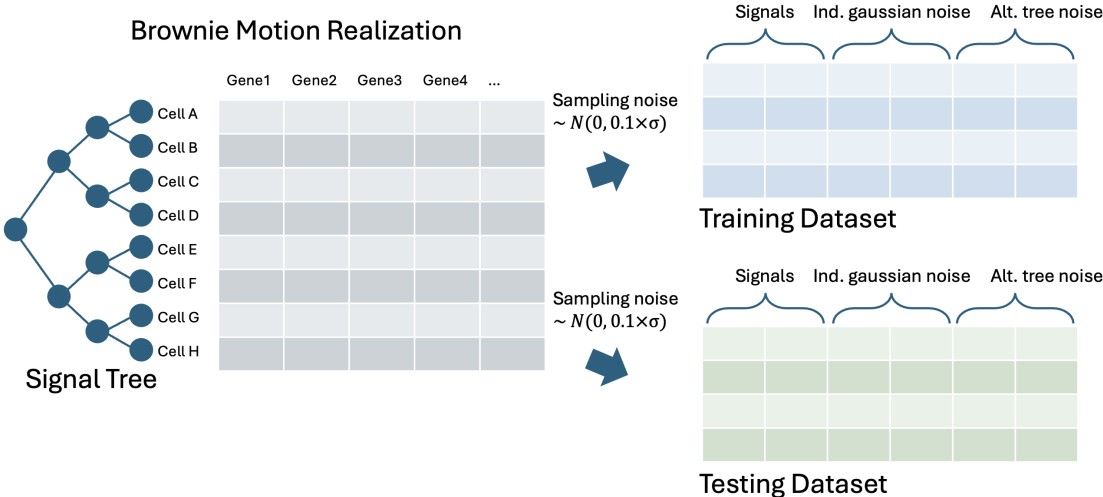

*Figure 10.* Illustration of the supervised setting for simulation. After generating signal variables through Brownian motion on the signal tree, a small amount of Gaussian noise is added to create a replicate set of signal features. One set is used for training, and the other for testing. Both the training and testing datasets then receive independent additions of Gaussian noise and alternative tree noise.

not pessimistic about finding some heuristic solutions under some reasonable constraints. After all, the very essence of systematics is to address this type of challenge.

A notable strength of our framework (Fig.2) is its ability to explicitly quantify the amount of prior topological information through the number of quartets with known structures. In a fully supervised setting, we assume complete knowledge of tree branch lengths and exact pairwise path lengths between terminal nodes. This scenario, while idealized, sets a performance upper bound for our model. Transitioning to weak supervision, we relax these requirements, retaining only knowledge of quartet topology (i.e., the relative order of distance sums) without branch length information. This allows optimization based on marginal properties of the embedding—specifically, quartet constraints.

Yet, the assumption of complete quartet topology knowledge remains stringent and rarely achievable in practice. In biological systems, partial qualitative information often exists that can group entities into distinct clades or categories. (Cavalli-Sforza & Edwards, 1967) For example, binary characters like vertebrate-invertebrate distinctions among animals or surface markers delineating immune cells. It is not uncommon to expect that we know some coarse groupings about the data. By leveraging these groupings, we can infer the topology of selected quartets. For instance, choosing two nodes from one clade and two from another ensures a butterfly configuration between the clades.

Thus, the quantity of known quartets can serve as a direct measure of prior information. In this work, we demonstrate that complete quartet topology enables accurate lineage reconstruction. Furthermore, when partial topology is available, we employ dictionary learning and quartet compatibility to iteratively expand the topology and refine the representation. This layered approach highlights the robustness of our framework across varying levels of supervision, paving the way for broader applications in cases where prior information is sparse or incomplete.

But still, knowing the topology of all the quartets is a stringent requirement that rarely is fulfilled in reality. However, in biology, we usually know some qualitative characters which can induce a partition of entities into distinct groups (Cavalli-Sforza & Edwards, 1967; Crowley et al., 2024). For example, the morphological feature vertebrate-invertebrate defines a binary (two-state) character. Some surface markers can be used to distinguish between broad lineage clades. We then can leverage grouping information by picking two nodes from one clade and picking two from another clade. The selected quartet must form into a butterfly bridge between the clades. We will have the topology of some of the quartets. So the number of known quartets can be quantitatively represented as the prior information. In this work, we set up the framework and showed that the problem is solvable if we know all the quartet topology. Then we show that when some of the quartets are known, we can use dictionary learning and quartet compatibility to learn the representation while expanding the topology information.

An intermediate scenario involves having only partial or grouped prior knowledge—such as knowing that certain leaves belong to a few large clades without having complete quartet-level constraints. In this case, we can initially focus on quartets formed from these known clades, where intra-clade topologies are more likely to be stable.

### E.2. Cell Aggregation

Another challenge we have set aside so far is defining an appropriate representation for each lineage node. In Figure 1, we presuppose that a single phenotype vector is readily available for every cell type. However, such an assumption glosses over a key biological complexity: the definition of cell types remains contentious. The rapid expansion of single-cell technologies [1] has vastly improved our capacity to measure cellular heterogeneity at scale, yet cell-type nomenclature often relies on a few canonical markers (Arendt et al., 2016)—a practice still widely debated (Domcke & Shendure, 2023).

From a developmental perspective, "lineage" offers a more objective concept: it captures the ground-truth history and potential future of a cell, quantifying how individual cells emerge and give rise to others within the same organism. Unfortunately, in most animals, obtaining a complete lineage-based taxonomy is not feasible. Visual tracking of cell divisions is only possible for transparent organisms such as the nematode *Caenorhabditis elegans*, where every cell division and fate has been cataloged in a single canonical lineage (Packer et al., 2019). This near-complete knowledge also extends to the closely related species, *C. briggsae*, which also displays a nearly invariant lineage (Large et al., 2024).

In our experiments, we leverage precisely these lineage-annotated single-cell RNA-seq datasets from (Packer et al., 2019) and (Large et al., 2024) as real-data benchmarks. Let $\{x^i\}_{i=1}^n$ denote the observed molecular profiles of $n$ individual cells, with each profile $x^i \in \mathbb{R}^p$. Suppose we have a mapping from cells to their respective lineage nodes (i.e., cell types). For lineage node $v$, let $S_v$ denote the subset of cells assigned to $v$. We then form an aggregate representation

$$x_v = \text{Aggregate}(x^i : i \in S_v),$$

where the aggregation could be as simple as computing the arithmetic mean if the cells exhibit minimal ambiguity in their type assignments. Indeed, in many practical single-cell atlas pipelines, a straightforward average of cells that share unambiguous lineage labels is often used (Packer et al., 2019; Large et al., 2024).

Nonetheless, complications arise because cell types are not always distinct clusters in high-dimensional space—individual cells may lie near the boundaries between nominally different types. Thus, assigning cells to discrete categories can be nontrivial, especially as the number of cell types grows. An alternative view is to treat cell aggregation as a set-based or multiple-instance problem (Kosiorek et al., 2019; Lee et al., 2019; Ilse et al., 2018; Zhao et al., 2021), in which each "bag" (set of cells) is mapped to a single label (the lineage node). Such methods provide flexible strategies for aggregating multiple cell profiles, accommodating overlap and uncertainty in cell-type definitions.

### E.3. Fraction of Unknown Quartets on Full Balanced Binary Tree

Suppose we have a full binary tree with $N$ leaves. Suppose we know the topology down to level $L$. So there are $k = 2^L$ clades at level L. Each clade has $n = \frac{N}{2^L}$ leaves. The number of all the quartets is

$$Q = \binom{n}{4} \approx \frac{N^4}{24}.$$

When $L = 1$, the fraction of known quartets is

$$Q_1 = \frac{\binom{n/2}{2}\binom{n/2}{2}}{\binom{n}{4}} = \frac{3\,n(n-2)}{8\,(n-1)(n-3)} \approx \frac{3n^2}{8n^2} = \frac{3}{8} = 0.375$$

So the fraction of unknown quartet $Q_0 = 1 - Q_1 = 0.625$.

---

[1] Nature Method of the Year 2013: single-cell sequencing; 2019: single-cell multimodal omics; 2020: spatially resolved transcriptomics.

When $L > 1$, the number of unknown quartets is:

$$Q_0 = \underbrace{\sum_{i=1}^{k} \binom{n}{4}}_{\text{(4)-type}} + \underbrace{\sum_{i=1}^{k} \binom{n}{3}(N-n)}_{\text{(3+1)-type}}$$

$$= k \binom{n}{4} + k \binom{n}{3}(N-n)$$

$$\approx 2^L \times \frac{n^4}{24} + 2^L \times \frac{n^3}{6}(N).$$

$$2^L \times \frac{n^4}{24} = 2^L \times \frac{(N/2^L)^4}{24} = \frac{N^4}{24}\frac{2^L}{(2^L)^4} = \frac{N^4}{24}\frac{1}{2^{3L}}.$$

$$2^L \times \frac{n^3}{6}N = 2^L \times \frac{(N/2^L)^3}{6}N = \frac{N^4}{6}\frac{2^L}{(2^L)^3} = \frac{N^4}{6}\frac{1}{2^{2L}}.$$

Therefore, the fraction of unknown quartets is roughly

$$Q_0 = \frac{\frac{N^4}{24}\frac{1}{2^{3L}} + \frac{4\,N^4}{24}\frac{1}{2^{2L}}}{\frac{N^4}{24}} = \frac{1}{2^{3L}} + \frac{4}{2^{2L}} = \frac{1}{2^{3L}} + \frac{1}{2^{2L-2}}.$$

Therefore, the fraction of unknown quartets is independent to the size of the tree.

- $L = 1, Q_0 = 0.625.$

- $L = 2, Q_0 = 0.265.$

- $L = 3, Q_0 = 0.064.$

## F. Model

### F.1. Model Architecture

Our proposed framework, CellTreeQM, aims to learn embeddings from high-dimensional phenotypic data that facilitate phylogenetic reconstruction. To effectively learn relationships among cells, we use a sequence of Transformer encoder blocks as the backbone of the network, illustrated in Figure 11. Unlike classical Transformer models, we do not include positional encodings, as our input cells are not inherently ordered. Without positional constraints, the self-attention mechanism can focus purely on learning meaningful relationships based on the feature similarities between cells.

Moreover, we incorporate two types of dropout regularization: Data Dropout, which is applied in the attention encoder to prevent overfitting on input features; Metric Dropout, a dropout layer added after the network's output layer(Qian et al., 2014). Empirically, we found out that Metric Dropout improves the model's performance. We call the dropout layer after the output layer the metric dropout. The dropout in the attention encoder is called data dropout.

For comparison, we also implemented a CellTreeQM-FC model with straightforward feedforward architecture. It consists of a stack of eight fully connected layers, each with a hidden dimension of 1024. Nonlinear activation functions (ReLU) are applied between layers to introduce model capacity and expressiveness.

We compared the performance between the network with Transformer encoder as backbones and Fully connected layers as backbone on a real dataset under a supervised setting. The network with the attention module performs better than FC. This indicates that while this FC is simple and relatively fast, it learns the pairwise distances within each quartet without modeling the global pairwise relationship.

For leave permutation, we randomly shuffle the labels of leaves. For cell permutation, we randomly shuffle the rows within each column. For gene permutation, we randomly shuffle the columns within each row. Those configurations serve as null experiments to see the performance of the model when we break the original pattern in the data.

*Table 5.* RF distances for various datasets and embeddings. Lower values indicate trees that are more similar to the ground truth topology. Permutation experiments are conduced with CellTreeQM-transformer. Experiments on C.elegans Small are repeated 3 times. Values in parentheses are standard errors.

| | N leaves | Dir. Recon. | CellTreeQM-transformer | CellTreeQM-fc | Leaves Permutation | Cell Permutation | Gene Permutation |
|---|---|---|---|---|---|---|---|
| C.elegans Small | 102 | 0.929 | 0.246 (0.012) | 0.400 (0.026) | 0.510 (0.058) | 0.927 (0.014) | 0.937 (0.012) |
| C.elegans Large | 295 | 0.955 | 0.571 | 0.647 | 0.876 | 0.969 | 0.973 |

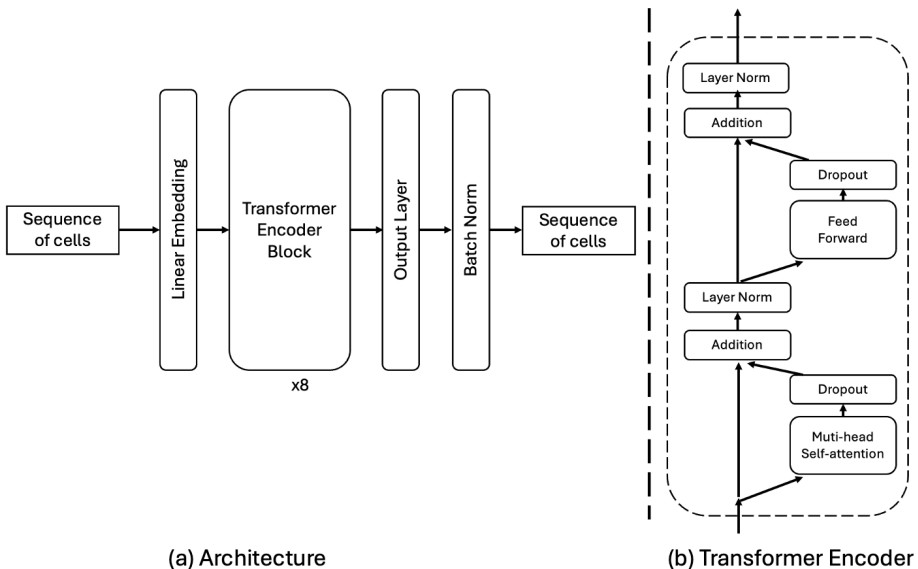

(a) Architecture  (b) Transformer Encoder

*Figure 11.* (a) The architecture of PhyloDist (b) Transformer Encoder Block

## F.2. Feature Gating

While enforcing an additive structure and preserving original distances remain central to our approach, many high-dimensional phenotypic datasets (e.g., scRNA-seq) include numerous features that do not reflect lineage-related variation. To address this, we introduce a *feature gating* module, which adaptively modulates the contribution of each input feature. By emphasizing lineage-relevant signals and down-weighting confounding or redundant attributes, the gating module can improve downstream tree reconstruction. To learn an effective feature mask, we use a Gumbel-Softmax (Gumbel, 1954) approach that promotes discrete gating decisions. Applying the gating is a simple elementwise product: $\tilde{x}_i = x_i \cdot g_i$.

**Gumbel Gating.** To learn an effective feature mask, we use a Gumbel-Softmax (Gumbel, 1954) approach that promotes discrete gating decisions:

$$g = \mathrm{GumbelSoftmax}\big(\mathrm{MLP}(E), \tau, \mathrm{hard} = \mathrm{True}\big),$$

where $E$ is an embedding for each feature, MLP projects this embedding into 2-dimensional logits $\ell_i$ (one for "off" and one for "on"), and $\tau$ is a temperature parameter. In the hard-sampling regime (hard = True), the gating weights become nearly binary (0 or 1), effectively pruning or retaining each feature. Applying the gating is a simple elementwise product: $\tilde{x}_i = x_i \cdot g_i$.

**Sparsity Penalty.** To encourage gating out superfluous features, we introduce a *sparsity penalty*:

$$\Omega_{\mathrm{gates}}^{(\mathrm{sparsity})} = \lambda_{\mathrm{spar}} \frac{\sum_i g_i}{\mathrm{input\ dim}},$$

where $\lambda_{\mathrm{spar}}$ is a tunable coefficient. Lower total activation ($\sum_i g_i$) results in a smaller penalty, thus incentivizing the network to keep fewer features "on." This penalty helps reduce noise and highlight relevant signals for lineage reconstruction.

## F.3. Loss Function Component Study

To further dissect the contributions of each component in our framework, we conduct an ablation study using the C.elegans Small dataset under a supervised setting. Table 6 reports the RF distances from trees reconstructed under various modified configurations of CellTreeQM, isolating the effects of the "close" and "push" terms of the quartet loss, as well as the regularization term $\Omega$.

The ablation results confirm that each component of our loss function plays a significant role in shaping the latent space. The combination of the "close" and "push" quartet constraints is essential to effectively approximate an additive metric, while the $\Omega$ regularization ensures that the learned representation remains grounded in the original phenotypic data. Together, these components enable CellTreeQM to achieve superior topology recovery compared to baseline or partially ablated models.

*Table 6.* Loss component study on the C.elegans-Dev dataset. Lower RF distance values indicate better topological similarity to the ground truth. The reported values are means across ten runs, with standard deviations in parentheses.

| | | Quartet Sampling | Deviation | Additivity | Close | Push | margin | Relative RF ↓ |
|---|---|---|---|---|---|---|---|---|
| baseline | | mismatched | 0.01 | 2 | 1 | 10 | 0.5 | 0.274 (0.021) |
| (a) | a1 | | 0 | | | | | 0.839 (0.011) |
| | a2 | | | 0 | | | | 0.846 (0.008) |
| (b) | b1 | | | 0.005 | | | | 0.838 (0.008) |
| | b2 | | | 0.01 | | | | 0.836 (0.008) |
| | b3 | | | 0.1 | | | | 0.707 (0.006) |
| | b4 | | | 0.5 | | | | 0.277 (0.016) |
| | b5 | | | 1 | | | | 0.253 (0.016) |
| | b6 | | | 4 | | | | 0.525 (0.203) |
| (d) | d1 | | | | 1 | 0 | | 0.514 (0.042) |
| | d2 | | | | 0 | 10 | | 0.547 (0.019) |
| (e) | e1 | | | | | 1 | | 0.433 (0.031) |
| | e2 | | | | | 2 | | 0.377 (0.026) |
| | e3 | | | | | 5 | | 0.321 (0.023) |
| | e4 | | | | | 20 | | 0.249 (0.019) |
| | e5 | | | | | 30 | | 0.257 (0.017) |
| (f) | f1 | | | | | | 0 | 0.267 (0.012) |
| | f2 | | | | | | 0.1 | 0.263 (0.017) |
| | f3 | | | | | | 0.3 | 0.275 (0.030) |
| | f4 | | | | | | 0.7 | 0.287 (0.040) |
| | f5 | | | | | | 1 | 0.314 (0.018) |
| (g) | g1 | matched | | | | | | 0.860 (0.018) |
| | g2 | all | | | | | | 0.306 (0.022) |

# G. Results

## G.1. Baselines

**Triplet Loss**  For each quartet of leaves, we first identify the two leaves with the smallest ground-truth distance (based on the known quartet ordering) and treat them as the *anchor* ($A$) and *positive* ($P$). We then select the leaf that is farthest from A to serve as the *negative* (N). Denoting the learned embeddings as $f(\cdot)$ and using a margin $m$, the triplet loss is defined as:

$$\mathcal{L}_{\text{tri}} = \sum_{(A,P,N)} \Big[ \|f(A) - f(P)\|^2 - \|f(A) - f(N)\|^2 + m_0 \Big]_+, \tag{1}$$

where $[\cdot]_+$ denotes the hinge function $\max(0, \cdot)$. Each quartet thus provides local distance orderings that the learned embedding space must respect.

**Quadruplet Loss**  We extend the triplet formulation by incorporating a second negative. From each quartet, after we identify the closest pair $(A, P)$, we designate *two* negatives, $N$ and $N'$, chosen among the more-distant leaves. One common

form of the quadruplet loss is:

$$\mathcal{L}_{\text{quad}} = \sum_{(A,P,N,N')} \left[ \|f(A) - f(P)\|^2 - \|f(A) - f(N)\|^2 + \alpha \right]_+ \tag{2}$$
$$+ \left[ \|f(A) - f(P)\|^2 - \|f(N') - f(N)\|^2 + \beta \right]_+.$$

where $\alpha$ and $\beta$ are margins. The first bracket encourages the anchor–positive distance to be smaller than the anchor–negative distance, as in triplet loss, while the second bracket enforces additional separation between $(A, P)$ and the second negative pair $(N', N)$, thereby enhancing global distance structure.

*CellTreeQM* and both baselines produce a learned embedding space from which we extract pairwise distances between leaves.

## G.2. Evaluation

We assess the reconstructed tree $\hat{\mathcal{T}}$ against a ground-truth tree $\mathcal{T}$ using metrics such as:

**Robinson–Foulds (RF) Distance.** The RF distance quantifies the topological difference between two unrooted trees by comparing their sets of partitions, where a partition corresponds to a split in the tree that divides the taxa into two complementary subsets. The metric counts how many partitions differ between the inferred and the true trees. We will generally present normalized RF distance, where 0 implies identical tree topology and 1 implies that no partitions are shared between the two trees.

**Quartet Distance (QD).** The RF distance is widely used due to its conceptual simplicity and biologist's focus on lineage relationships (*i.e.*, tree topology). However, it does have limitations. Because it treats all partitions equally, it does not distinguish between topological differences that might have different biological relevance. More importantly, the distance measure is sensitive to tree differences that arise by a cut-and-paste operation. For example, if one leaf vertex is moved from one side of the tree to another, RF might be 1, even if the remaining tree structure is identical. Thus, to consider more detailed subtree structure, we introduce quartet distance(Bryant et al., 2000). Given any four leaf vertices (*i.e.*, a quartet of leaves), there are three possible arrangements of their unrooted tree-graph relationships. Given two tree graphs, we consider all possible quartets of leaves and compute the percent of quartet tree graphs that are different as the quartet distance. For large trees, we may approximate the quartet distance by a sample of the leaf vertices.

For reference, we also compare to a "direct reconstruction" approach that uses raw Euclidean distances in gene-expression space, denoted as base RF and base QD. We use $\Delta\%$RF and $\Delta\%$QD to measure the relative improvement in RF or QD over direct raw-data reconstruction (higher is better).

$$\Delta\%\text{RF} = \frac{\text{RF}_{\text{base}} - \text{RF}_{\text{recon}}}{\text{RF}_{\text{base}}}; \qquad \Delta\%\text{QD} = \frac{\text{QD}_{\text{base}} - \text{QD}_{\text{recon}}}{\text{QD}_{\text{base}}} \tag{3}$$

**Label Permutation as Null Experiments** To verify that our method is genuinely learning lineage information (rather than artifacts of high-dimensional data), we permute the mapping of data rows to lineage leaves, effectively scrambling the ground-truth relationships as leaves permutation. The permutations break the biological (or simulated) structure that underlies the lineage tree.

## G.3. Supervised Setting

**Simulation** Table 7 summarizes the performance of our method under four simulated scenarios (A–D), each parameterized by distinct signal, noise, and alternate tree noise settings. In each scenario, we vary the dimensionality of the signal features ($d_{\text{sig}}$), Gaussian noise ($d_{\text{noise}}$), and alternate tree noise ($d_{\text{AltSig}}$), as well as the scaling factors $\alpha$ (for noise standard deviation) and $\beta$ (for branch lengths in the alternate tree). We define the signal-to-noise ratio (SNR) by the ratio of the total signal dimensionality to the sum of noise and alternate tree noise dimensions, $\text{SNR} = d_{\text{sig}}/(d_{\text{noise}} + T \times d_{\text{AltSig}})$. The total feature dimensionality d thus reflects both signal and noise features combined.

Under "CellTreeQM" we compare two modes: $\mathcal{F}$ (using all features) and $\mathcal{G}$ (applying a Gumbel-based feature gate). The final three columns ("Recall," "Precis.," $\Delta$) further illustrate how well the gating module identifies true signal features. We observe that using all features without gating ($\mathcal{F}$) can be prone to noise contamination when $d_{\text{noise}}$ and $d_{\text{AltSig}}$ are large, while the Gumbel-based gate ($\mathcal{G}$) achieves higher Recall and Precision of signal features and remains relatively stable (as seen in

$\Delta$). These trends hold across the different scenarios, with certain variations in performance as the SNR, scaling factors $\alpha$ and $\beta$, and total dimensionality d change. For additional scenarios with $n_{\text{leaves}}$ = 128 and 256, as well as per-run standard deviations, we refer readers to the appendix.

*Table 7.* **Performance comparison on simulated data with $n_{\text{leaves}}$ = 64 (scenarios A–D).** Each row shows parameter choices for the signal ($w$, $d_{\text{sig}}$), Gaussian noise ($d_{\text{noise}}$, $\alpha$), and alternate tree noise ($S$, $T$, $d_{\text{AltSig}}$, $\beta$). $\alpha$ is the scaling factor of noise stander deviation such that $\sigma_{noise} = \alpha\bar{\sigma}_{sig}$. $\beta$ is the scaling factor of the branch length range of the alternative tree such that $w_{\text{alt}} = \beta w$. We define SNR as SNR = $d_{\text{sig}}/(d_{\text{noise}} + T \times d_{\text{AltSig}})$ Here, $d$ denotes the total feature dimensionality. "Direct Recon." measures reconstruction when using only the signal features (Signal) vs. when using both signal and noise (Noisy). "CellTreeQM" ($\mathcal{F}, \mathcal{G}$) indicates whether all features are used ($\mathcal{F}$) or a Gumbel-based feature gate is applied ($\mathcal{G}$). Each value is averaged over five runs. "$\mathcal{G}$-Gate Performance" columns show: (*Recall*) the percentage of signal features selected (out of all signal features) at the best-performing training step; (*Precis.*) the percentage of selected features that are signal features at that same step; ($\Delta$ g) the standard deviation of the number of gate changes over the final one-third of training steps. Each value is averaged over five runs. Additional results (including standard deviations) for $n_{\text{leaves}}$ = 128 and 256 are provided in the appendix.

| | | Signal | Gauss. Noise | | Alt. Tree Noise | | | | Summary | | Direct Recon. | | CellTreeQM | | $\mathcal{G}$-Gate Performance | | |
|---|---|---|---|---|---|---|---|---|---|---|---|---|---|---|---|---|---|
| | $w$ | $d_{\text{sig}}$ | $d_{\text{noise}}$ | $\alpha$ | $S$ | $T$ | $d_{\text{AltSig}}$ | $\beta$ | SNR | $d$ | Signal | Noisy | $\mathcal{F}$ | $\mathcal{G}$ | Recall | Precis. | $\Delta g$ |
| | 2 | 20 | 20 | 0.5 | 20 | 1 | 20 | 0.5 | 0.5 | 60 | 0.291 | 0.615 | 0.318 | 0.126 | 0.85 | 0.946 | 1.276 |
| A | 2 | 50 | 50 | 0.5 | 50 | 1 | 50 | 0.5 | 0.5 | 150 | 0.095 | 0.413 | 0.103 | 0.041 | 0.804 | 0.951 | 2.158 |
| | 2 | 100 | 100 | 0.5 | 100 | 1 | 100 | 0.5 | 0.5 | 300 | 0.028 | 0.275 | 0.037 | 0.014 | 0.824 | 0.884 | 3.624 |
| | 2 | 20 | 100 | 0.5 | | | | | 0.2 | 120 | 0.292 | 0.534 | 0.512 | 0.182 | 0.94 | 0.775 | 3.064 |
| B | 5 | 50 | 500 | 1 | | | | | 0.1 | 550 | 0.156 | 0.865 | 0.884 | 0.370 | 0.9 | 0.424 | 17.267 |
| | 10 | 100 | 2000 | 2 | | | | | 0.05 | 2100 | 0.157 | 0.980 | 0.960 | 0.867 | 0.794 | 0.130 | 105.705 |
| | 2 | 20 | | | 20 | 1 | 20 | 0.5 | 1 | 40 | 0.315 | 0.567 | 0.233 | 0.108 | 0.89 | 0.981 | 1.217 |
| C | 5 | 50 | | | 50 | 2 | 100 | 1 | 0.5 | 150 | 0.177 | 0.951 | 0.603 | 0.056 | 0.848 | 0.954 | 3.605 |
| | 10 | 100 | | | 100 | 2 | 200 | 2 | 0.5 | 300 | 0.144 | 0.997 | 0.856 | 0.263 | 0.844 | 0.738 | 12.961 |
| | 2 | 20 | 100 | 0.5 | 20 | 1 | 20 | 0.5 | 0.17 | 140 | 0.308 | 0.663 | 0.600 | 0.249 | 0.98 | 0.663 | 3.407 |
| D | 5 | 50 | 500 | 1 | 50 | 2 | 100 | 1 | 0.08 | 650 | 0.174 | 0.948 | 0.921 | 0.403 | 0.828 | 0.403 | 22.853 |
| | 10 | 100 | 2000 | 2 | 100 | 2 | 200 | 2 | 0.05 | 2300 | 0.144 | 0.984 | 0.974 | 0.875 | 0.772 | 0.112 | 129.442 |

**Real Data** Results for the supervised setting on all real datasets are summarized in Table 8. For the *C. elegans* Small and Mid datasets, we report the mean and standard deviation over three independent runs. For the *C. elegans Large* dataset, we report results from a single run due to the high computational cost—stemming from both the large number of training quartets (corresponding to 295 leaves) and the inefficient implementation of RF distance used during evaluation.

**Permutation experiments validate quartet-based fitting.** Training *CellTreeQM* on a label permuted tree $T'$ still embeds the data with moderate fidelity ($\Delta\%$RF), reflecting information in an unlabeled tree, but less accurately than with the true tree. In contrast, triplet and quadruplet models struggle to align a random tree with the data. Similar trends hold for the *C. elegans Mid* and *Large* datasets, where triplet/quadruplet losses improve over raw data but still lag behind *CellTreeQM* (see Table 8). These results echo the recent findings in (Schlüter & Uhler, 2025) showing that *true lineage topologies* yield lower training losses and more generalizable embeddings, thus confirming there is ample lineage signal in the gene expression data—our method is not merely overfitting.

**Visualization on *C. elegans* Small** CellTreeQM embeds the leaves into 128-dimensional space. To have some intuition about the latent space, we project the embeddings to 20 principal components (via PCA), followed by a 2D t-SNE projection. Figure 12 displays the results on *C. elegans Small*, coloring cells by common ancestors at different hierarchical levels. While triplet and quadruplet losses capture some broad structure, *CellTreeQM* more effectively organizes the leaves according to the underlying tree hierarchy.

In addition to these embeddings, Figure 13 illustrates the final lineage trees reconstructed from the learned distances of *CellTreeQM*, Triplet, and Quadruplet. Each leaf is colored according to its major lineage annotation in *C. elegans*. Notably, *CellTreeQM* yields a more faithful global topology, with tighter, lineage-consistent subtrees. By contrast, trees reconstructed under triplet or quadruplet losses exhibit more dispersed leaf placements, sometimes grouping distantly related lineages together. These visual differences align with our quantitative findings in Table 1 and underscore *CellTreeQM*'s advantage in accurately capturing the hierarchical relationships among cells.

**Reconstruction methods** To evaluate the learned embeddings across different tree reconstruction algorithms, we compared five methods—Neighbor-Joining (NJ), UPGMA, FastME, Ward, and Single linkage—on the *C. elegans* Small dataset.

Across all five reconstruction methods, CellTreeQM consistently achieves the lowest RF distances on both training and testing sets, demonstrating robust learning and generalization capabilities. As shown in Figure 14, the radar plot highlights the relative RF improvement of CellTreeQM compared to models trained with Triplet and Quadruplet losses across all reconstruction methods.

Interestingly, the Ward method slightly outperforms NJ in some settings, despite NJ being theoretically optimal for additive trees. This may suggest that our learned embeddings support variance-based clustering more effectively in certain scenarios. We leave a more detailed analysis of this behavior to future work.

### G.4. Weakly Supervised Setting

#### G.4.1. HIGH LEVEL PARTITION SETTING.

We report the *known-quartet distance* (K-QDist) and the *unknown-quartet distance* (U-QDist). Unsurprisingly, K-QDist remains near zero for successful methods, since those quartets are directly supervised. Meanwhile, U-QDist steadily decreases at deeper prior levels, indicating that CellTreeQM *generalizes* better fits unconstrained quartets as more high-level clades become known. We also measure $\Delta\text{RF} = \text{RF0} - \text{RF}$, the improvement over a raw-data baseline. This gap grows with increased prior, demonstrating that additional top-level constraints help the model recover more correct branching patterns.

To gauge how well the model recovers the unknown internal structure *within* each clade, we also track local subtree errors at various "levels" of the reconstructed tree. Even though these subclades were not all explicitly supervised, the table reveals that CellTreeQM obtains lower Robinson–Foulds distances on these deeper substructures—particularly in the presence of more known quartets.

We report the fraction of known quartets at levels of the full balanced dataset with 64 leaves and for the three *C. elegans* datasets in Table 9.

**Performance on Simulated Data.**  The results for the high-level partition setting are shown in Figure 15. The dataset is simulated with $n_{\text{leaves}} = 64$ and a maximum walk of 2 per branch. Each leaf contains 50 signal features and 500 Gaussian noise features, with the same mean and standard deviation as the signal features. Additionally, each leaf has 50 spurious features derived from an alternative tree. There are two alternative trees, each with 32 leaves and the same maximum walk as the true tree. Each experiment is repeated 10 times, and the mean and standard deviation are shown as the central value and error bars in the figure.

**Performance on *C. elegans* Data.**  Under the high-level partition setting, the results for *C. elegans* Small are shown in Figure 20 and Table 10. The results for *C. elegans* Mid are shown in Figure 21 and Table 11. The results for *C. elegans* Large are shown in Figure 21 and Table 12. The experiments for *C. elegans* Small are repeated 10 times. The experiments for *C. elegans* Mid are repeated 3 times. The experiments for *C. elegans* Large are run once due to the high computational cost.

#### G.4.2. PARTIAL LABELED LEAVES SETTING.

In many biological systems, researchers may know a coarse-grained lineage structure up to a certain "level" of the tree but not the fine-grained branching events beneath it. Concretely, let us define a level as the number of branching steps from the root. We assume we know how leaves are divided into clades at level, as well as the relationships among these clades (see Fig. 2, left). Hence, each leaf is assigned to exactly one of these high-level clades, but the tree structure *within* each clade remains unknown.

When forming quartets under this partial knowledge, some quartets still have ambiguous structure ("unknown quartets"). Specifically, a quartet is unknown if all four leaves lie in the same clade or if three of the leaves come from the same clade, because the high-level partition does not determine the precise branching among these leaves. Quartets with leaves spanning different clades (e.g., two leaves from one clade and two from another) become "known quartets," whose topological order *can* be inferred from the clade-level prior. In our weakly supervised training, we compute the additivity loss *only* on these known quartets, since their correct topology is implied by the high-level partitions.

The experiments for *C. elegans* Small are repeated 10 times. The experiments for *C. elegans* Mid are repeated 5 times. We did not run any experiment on *C.elegans* Large because the results on *C. elegans* Small and Mid. demonstrate the difficulty

of this setting. Table 13 reports the Known, Partial, and Unknown counts and proportions for different known fractions in *C. elegans* Small and Mid. datasets.

**Performance on Simulated Data.** The results for the partial-labeled setting are shown in Figure 19 with known fractions as $35\%, 55\%, 85\%$. The dataset is simulated with $n_{\text{leaves}} = 128$ and a maximum walk of 2 per branch. Each leaf contains 50 signal features and 500 Gaussian noise features, with the same mean and standard deviation as the signal features. Additionally, each leaf has 50 spurious features derived from an alternative tree. There are two alternative trees, each with 64 leaves and the same maximum walk as the true tree. Each experiment is repeated 10 times, and the mean and standard deviation are shown as the central value and error bars in the figure.

**Performance on *C. elegans* Data.** The results for C.elegans Small and Mid are summarized in Table 14 and 15 for fraction $30\%, 50\%, 80\%$. The results are also visualized by bar plots in Figure. 20 and 21. The experiments for *C. elegans* Small are repeated 10 times. The experiments for *C. elegans* Mid are repeated 5 times.

### G.5. Unsupervised Setting

Here, we provide a preliminary exploration to assess whether the *CellTreeQM* framework can be extended to an unsupervised learning scenario. We use a Brownian motion simulation to generate a phylogenetic tree with 128 leaves with unit length. Each leaf is represented by 32 signal features and 32 Gaussian noise features, with the noise standard deviation set to 10. A direct reconstruction using the noisy data yields an RF value of 0.968. As baselines, we applied PCA with 5, 10, and 20 principal components, as well as Gaussian random projections with 2, 5, and 10 dimensions; all of these produced an RF value of 1. When we reconstruct the tree only based on the signal features, we got the optimal reconstruction accuracy RF = 0.128.

To adapt *CellTreeQM* to the unsupervised setting, we make two main modifications to simplify the model compared to its supervised and weakly supervised versions:

1. We replace transformer encoder a fully connected backbone of 3 layers, each with 32 hidden units.

2. We replace the Gumbel-Softmax feature gating with a learnable linear projection matrix $G \in \mathbb{R}^{d \times d}$, where d is the number of input features.

Because we do not have ground-truth distances among leaves in the unsupervised scenario, we approximate additivity by sorting the quartet distances $\{S_1, S_2, S_3\}$ (see 5.1) based on their values in the embedding space. Following the same idea as in DeSeto's algorithm, we define the "close" term as: and take

$$\mathcal{L}_{\text{quartet}} = \mathcal{L}_{\text{close}}, \quad \mathcal{L}_{\text{additivity}} = \frac{1}{Q} \sum_{q=1}^{Q} \mathcal{L}_{\text{quartet}},$$

where Q is the total number of quartets.

We regularize the projection matrix G to encourage excluding noise features without excessively penalizing omissions. Specifically, we initialize G as the identity matrix and apply an $L_1$ penalty $\|G - I\|_1$. Note that, unlike in the supervised and weakly supervised settings—where feature gating is applied directly to the input—here we apply feature gating only to the *deviation* component. Concretely, we compare the learned embedding f(X) to its linear projection $GX$ via

$$\Omega(f, X) = \frac{1}{N} \big\| \mathcal{D}\big(f(X)\big) - \mathcal{D}\big(GX\big) \big\|_F^2,$$

where $\mathcal{D}$ is an operator (e.g., pairwise distance) applied to the corresponding space. The overall loss function then combines the additivity loss, the embedding deviation term, and the regularization on G:

$$\min_{f,G} \Big[ \mathcal{L}_{\text{additivity}}(f, X) \; + \; \lambda \Omega(f, X) \; + \; \big\| G - I \big\|_1. \Big]$$

Figure 22 illustrates how *CellTreeQM* learns in a purely unsupervised setting over the course of training on a simulated dataset. The figure shows that *CellTreeQM*, even without any labels or supervision, progressively approaches the signal-only

baseline, suggesting that the model's gating mechanism and the unsupervised objective are successfully filtering out noise and capturing the underlying signal.

## H. Training Setups

We evaluated our method in two settings: *supervised* and *weakly supervised*. For both settings, we tested on simulated and real datasets. Unless otherwise noted, we did not extensively tune the hyperparameters.

**Supervised Setting.**

- **Simulation:** We use a Transformer with 8 layers and 2 attention heads. The projection layer and hidden layer are both 256-dimensional, and the model outputs 128-dimensional embeddings. We apply a data dropout of 0.3 and a metric dropout of 0.2. When necessary, we set the gate regularization weight to 5.

- **Real data:** The Transformer also has 8 layers and 2 attention heads, but with 1024-dimensional projection and hidden layers, producing 128-dimensional outputs. Here, we use a data dropout of 0.1 and a metric dropout of 0.1. When needed, the gate regularization weight is 0.01.

**Weakly Supervised Setting.**

- **Simulation:** We use a Transformer with 4 layers and 2 attention heads. The projection and hidden layers are both 256-dimensional, and the model outputs 128-dimensional embeddings. We apply a data dropout of 0.3 and a metric dropout of 0.2. When required, the gate regularization weight is 8.

- **Real data:** The Transformer has 8 layers and 2 attention heads, with projection and hidden layers of 256 dimensions, producing 128-dimensional outputs. We use a data dropout of 0.3 and a metric dropout of 0.2, and set the gate regularization weight to 0.01 when needed.

## I. More Results

In the main text, we demonstrated that CellTreeQM considerably improves lineage reconstruction accuracy over standard contrastive baselines on *C.elegans* datasets. We also showed that the *high-level partition* setting is a practical and effective form of weak supervision, whereas the *partially leaf-labeled* setting remains too challenging to be useful in practice.

To further validate the generalizability and discovery potential of CellTreeQM, we evaluated the model on two additional CRISPR-based lineage datasets derived from mouse embryonic stem cell (mESC) clone 1D5 with non-targeting sgRNA: 3435_NT_T1 (151 cells) and 3435_NT_T6 (91 cells), obtained from Chan et al. (2019). Proxy ground truth trees were reconstructed using Cassiopeia. Due to ambiguity in the tree structure, we sampled binary trees from the full lineage. Quantitative results are shown in Table 16, 17. The corresponding barplots are shown in figure 23 and 24.

These experiments reinforce our findings on the *C. elegans* data: CellTreeQM consistently outperforms baseline methods, particularly in generalizing to unseen quartets. We applied CellTreeQM without dataset-specific tuning. We expect that further gains may be possible with targeted optimization.

*Table 8.* **Supervised results on Cell Lineage Benchmark.** Direct reconstruction on raw data yields Base RF and Base QDist. Suffix "-G" denotes feature gating, and "-p" indicates label permutation. Standard deviations are in parentheses. Experiments are repeat 3 times expect C. elegans Large.

| Method | Train RF↓ | Test RF↓ | Test QD↓ | Δ%RF↑ | Δ%QD↑ |
|---|---|---|---|---|---|
| **C. elegans Small**, Base RF=0.923; Base QD=0.554 | | | | | |
| CellTreeQM | **0.000** (0.00) | **0.286** (0.05) | **0.074** (0.01) | **0.690** (0.05) | **0.867** (0.02) |
| CellTreeQM-G | **0.000** (0.00) | **0.226** (0.02) | **0.084** (0.01) | **0.757** (0.03) | **0.848** (0.01) |
| CellTreeQM-p | 0.013 (0.00) | 0.566 (0.04) | 0.208 (0.03) | 0.434 (0.04) | 0.691 (0.05) |
| Triplet | 0.519 (0.03) | 0.741 (0.01) | 0.201 (0.01) | 0.179 (0.01) | 0.637 (0.02) |
| Triplet-G | 0.545 (0.03) | 0.724 (0.00) | 0.207 (0.01) | 0.203 (0.03) | 0.631 (0.02) |
| Triplet-p | 0.677 (0.06) | 0.963 (0.01) | 0.413 (0.04) | 0.037 (0.01) | 0.385 (0.06) |
| Quadruplet | 0.057 (0.00) | 0.492 (0.03) | 0.120 (0.00) | 0.454 (0.02) | 0.784 (0.01) |
| Quadruplet-G | 0.061 (0.00) | 0.471 (0.04) | 0.116 (0.00) | 0.484 (0.04) | 0.791 (0.01) |
| Quadruplet-p | 0.118 (0.00) | 0.848 (0.01) | 0.310 (0.01) | 0.149 (0.01) | 0.538 (0.02) |
| **C. elegans Mid**, Base RF=0.967, Base QD=0.579 | | | | | |
| CellTreeQM | **0.022** (0.00) | **0.513** (0.01) | **0.155** (0.00) | **0.457** (0.02) | **0.730** (0.01) |
| CellTreeQM-G | **0.031** (0.01) | **0.472** (0.00) | **0.130** (0.01) | **0.490** (0.01) | **0.773** (0.01) |
| CellTreeQM-p | 0.035 (0.01) | 0.831 (0.00) | 0.356 (0.02) | 0.165 (0.00) | 0.466 (0.03) |
| Triplet | 0.650 (0.02) | 0.811 (0.00) | 0.189 (0.00) | 0.095 (0.01) | 0.673 (0.01) |
| Triplet-G | 0.693 (0.02) | 0.809 (0.01) | 0.183 (0.00) | 0.112 (0.01) | 0.682 (0.00) |
| Triplet-p | 0.878 (0.03) | 0.970 (0.01) | 0.438 (0.01) | 0.026 (0.01) | 0.344 (0.01) |
| Quadruplet | 0.146 (0.01) | 0.557 (0.01) | 0.179 (0.01) | 0.396 (0.01) | 0.690 (0.01) |
| Quadruplet-G | 0.174 (0.04) | 0.537 (0.01) | 0.166 (0.01) | 0.423 (0.02) | 0.711 (0.02) |
| Quadruplet-p | 0.283 (0.03) | 0.915 (0.01) | 0.395 (0.00) | 0.082 (0.01) | 0.409 (0.01) |
| **C. elegans Large**, Base RF=0.949; Base QD=0.586 | | | | | |
| CellTreeQM | **0.113** | **0.568** | **0.111** | **0.401** | **0.811** |
| CellTreeQM-G | **0.140** | **0.521** | **0.115** | **0.439** | **0.805** |
| CellTreeQM-p | 0.195 | 0.863 | 0.262 | 0.137 | 0.606 |
| Triplet | 0.822 | 0.863 | 0.229 | 0.077 | 0.608 |
| Triplet-G | 0.808 | 0.873 | 0.196 | 0.073 | 0.664 |
| Triplet-p | 0.979 | 0.993 | 0.424 | 0.007 | 0.365 |
| Quadruplet | 0.432 | 0.675 | 0.155 | 0.284 | 0.736 |
| Quadruplet-G | 0.377 | 0.712 | 0.144 | 0.249 | 0.754 |
| Quadruplet-p | 0.620 | 0.932 | 0.317 | 0.068 | 0.523 |

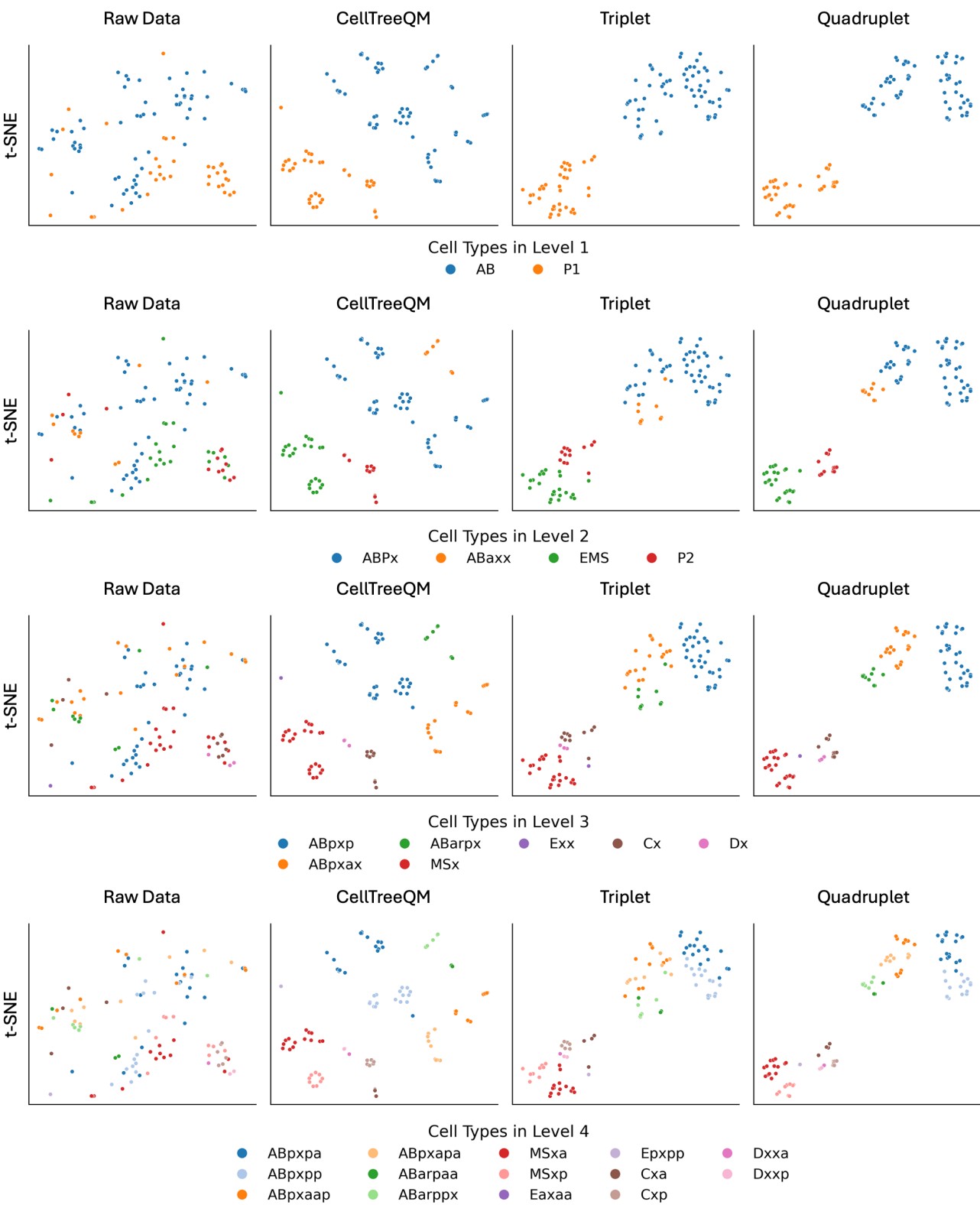

*Figure 12.* t-SNE visualization of CellTreeQM embeddings for *C. elegans Small*. The embeddings are first reduced to 20 principal components via PCA before applying t-SNE. Each panel corresponds to a different hierarchical level, with colors representing common ancestors at that level.

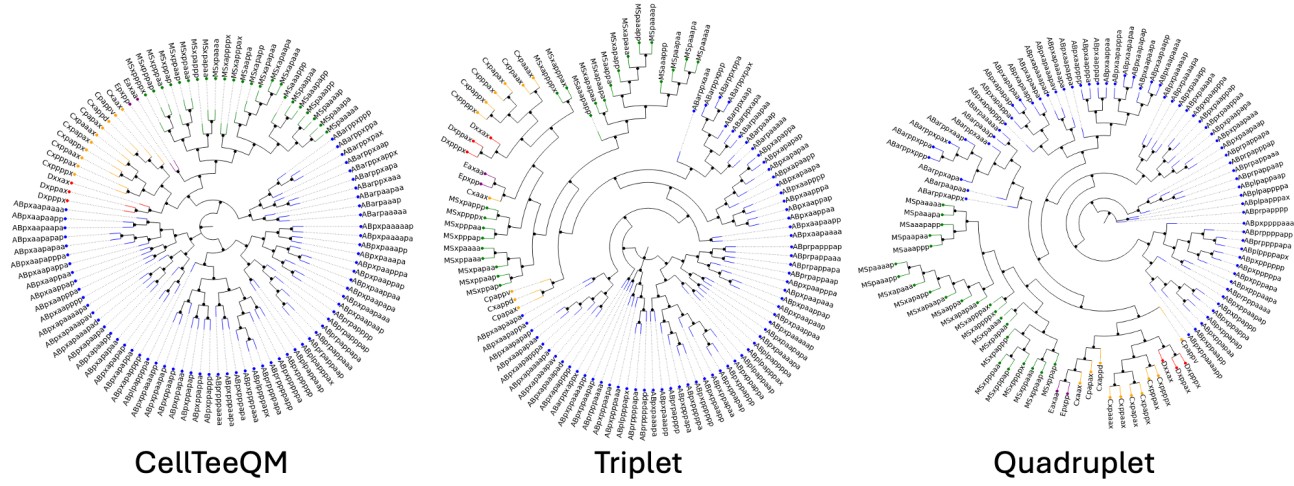

*Figure 13.* Reconstructed circular lineage tree for the *C. elegans* dataset using the learned embeddings from *CellTreeQM*. Each leaf is color-coded according to its major lineage (e.g., AB, MS, E, C, D), with the circular layout highlighting the hierarchical structure. The faithful grouping of leaves into cohesive subtrees demonstrates the effectiveness of *CellTreeQM* in capturing global genealogical relationships from the data.

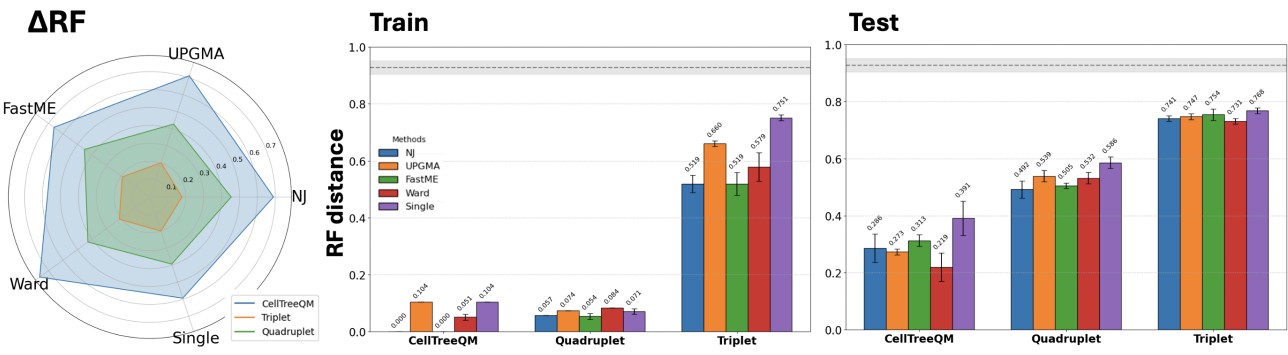

*Figure 14.* Comparison of reconstruction accuracy (measured by RF distance) across five tree-building algorithms. Left: $\Delta RF$ radar plot comparing CellTreeQM, Triplet, and Quadruplet models. Middle and Right: bar plots of RF distances on train and test sets, respectively.

*Table 9.* Fraction of Known Quartets at Levels.

|  | Brownian64 | | C. elegans Small | | C. elegans Mid | | C. elegans Large | |
|---|---|---|---|---|---|---|---|---|
| **n leaves** | 64 | | 102 | | 182 | | 295 | |
| **n quartets** | 635,376 | | 4,249,575 | | 44,224,635 | | 309,177,995 | |
|  | **Counts** | **Prop.** | **Counts** | **Prop.** | **Counts** | **Prop.** | **Counts** | **Prop.** |
| Level 1 | 238,266 | 0.375 | 1,417,233 | 0.3335 | 11,814,336 | 0.2613 | 77,232,663 | 0.2498 |
| Level 2 | 467,001 | 0.735 | 2,515,748 | 0.592 | 31,859,427 | 0.7204 | 220,351,157 | 0.7127 |
| Level 3 | 594,711 | 0.936 | 3,526,722 | 0.8299 | 39,815,438 | 0.9003 | 280,238,934 | 0.9064 |
| Level 4 | 635,376 | 1.000 | 4,057,919 | 0.9549 | 43,012,880 | 0.9726 | 301,634,051 | 0.9756 |

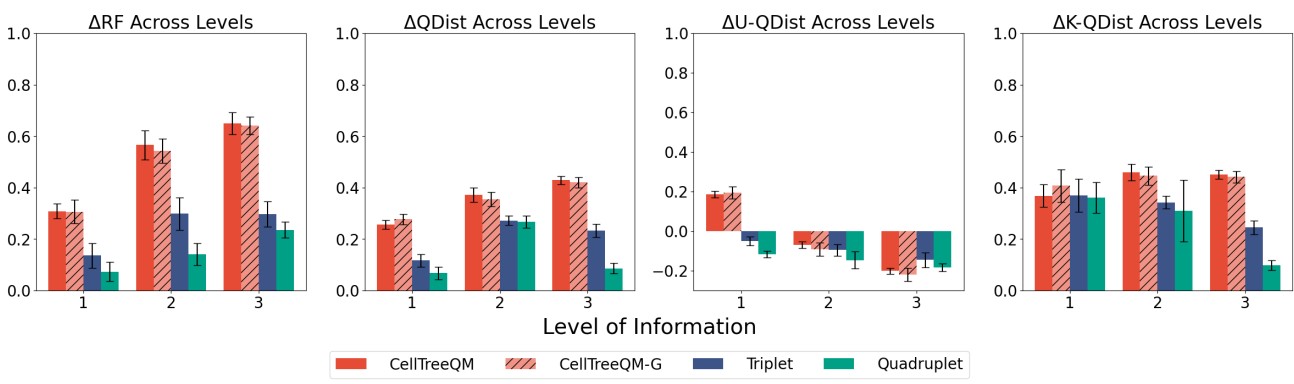

*Figure 15.* Results for **high-level partition** setting on simulation with 64 leaves.

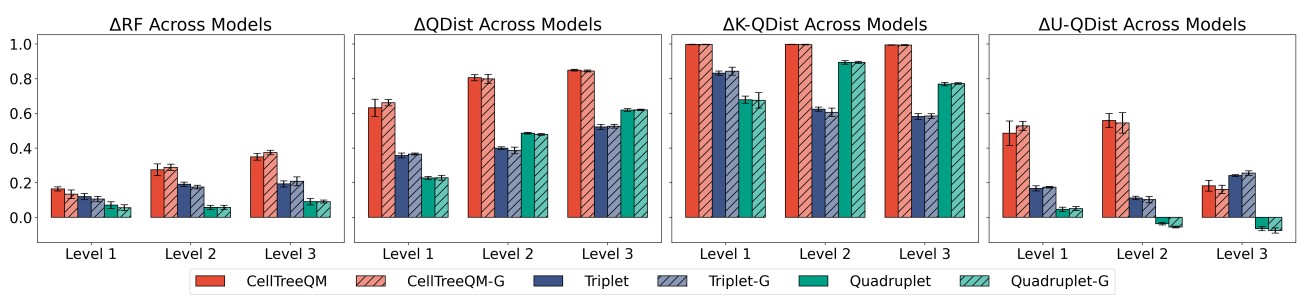

*Figure 16.* Results for **high-level partition** setting on C.elegans Small dataset. x-axis is the level of partition.

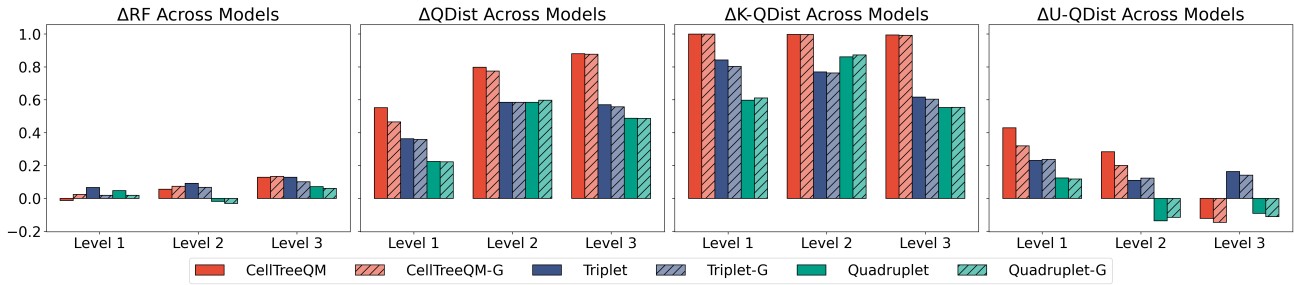

*Figure 17.* Results for **high-level partition** setting on C.elegans Mid dataset. x-axis is the level of partition.

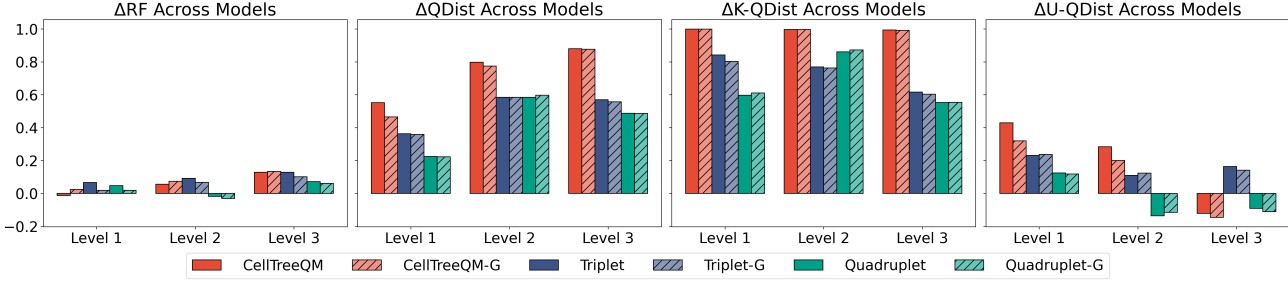

*Figure 18.* Results for **high-level partition** setting on C.elegans Large dataset. x-axis is the level of partition.

*Table 10.* **Weakly supervised High-level partition results on *C. elegans Small* under different partition levels.** K-QD and U-QD are quartet distances on the *known* and *unknown* quartets, respectively. he reported values are means across three runs, with standard deviations in parentheses.

| Method | RF↓ | △%RF↑ | QD↓ | △%QD↑ | △%K-QD↑ | △%U-QD↑ |
|---|---|---|---|---|---|---|
| **Partition Level: 3** | | | | | | |
| CellTreeQM | **0.592(0.02)** | **0.349(0.02)** | **0.083(0.00)** | **0.849(0.01)** | **0.994(0.00)** | 0.182(0.03) |
| CellTreeQM-G | 0.576(0.02) | **0.374(0.01)** | 0.086(0.00) | **0.844(0.01)** | **0.993(0.00)** | 0.161(0.02) |
| Triplet | 0.735(0.01) | 0.193(0.02) | 0.263(0.01) | 0.521(0.01) | 0.582(0.02) | **0.241(0.00)** |
| Triplet-G | **0.723(0.02)** | 0.208(0.03) | 0.260(0.01) | 0.526(0.01) | 0.584(0.01) | **0.255(0.01)** |
| Quadruplet | 0.832(0.02) | 0.090(0.02) | 0.209(0.00) | 0.619(0.01) | 0.768(0.01) | -0.065(0.01) |
| Quadruplet-G | 0.828(0.01) | 0.091(0.01) | 0.208(0.00) | 0.620(0.00) | 0.772(0.00) | -0.076(0.02) |
| **Partition Level: 2** | | | | | | |
| CellTreeQM | **0.657(0.02)** | **0.274(0.03)** | **0.107(0.01)** | **0.805(0.02)** | **0.998(0.00)** | **0.558(0.04)** |
| CellTreeQM-G | **0.653(0.02)** | **0.289(0.02)** | **0.110(0.01)** | **0.799(0.03)** | **0.996(0.00)** | **0.545(0.06)** |
| Triplet | 0.745(0.01) | 0.191(0.01) | 0.329(0.00) | 0.399(0.01) | 0.623(0.01) | 0.111(0.01) |
| Triplet-G | 0.756(0.01) | 0.174(0.01) | 0.337(0.01) | 0.386(0.02) | 0.606(0.02) | 0.102(0.02) |
| Quadruplet | 0.857(0.01) | 0.058(0.01) | 0.282(0.00) | 0.485(0.01) | 0.894(0.01) | -0.037(0.01) |
| Quadruplet-G | 0.863(0.00) | 0.057(0.01) | 0.287(0.00) | 0.478(0.01) | 0.894(0.01) | -0.056(0.01) |
| **Partition Level: 1** | | | | | | |
| CellTreeQM | **0.772(0.02)** | **0.164(0.01)** | **0.202(0.03)** | **0.631(0.05)** | **0.997(0.00)** | **0.485(0.07)** |
| CellTreeQM-G | **0.788(0.02)** | **0.133(0.02)** | **0.186(0.01)** | **0.661(0.02)** | **0.997(0.00)** | **0.527(0.03)** |
| Triplet | 0.814(0.02) | 0.120(0.02) | 0.353(0.01) | 0.356(0.01) | 0.832(0.01) | 0.167(0.02) |
| Triplet-G | 0.808(0.02) | 0.105(0.02) | 0.348(0.00) | 0.365(0.01) | 0.843(0.02) | 0.173(0.00) |
| Quadruplet | 0.853(0.01) | 0.070(0.02) | 0.424(0.00) | 0.227(0.01) | 0.678(0.02) | 0.046(0.01) |
| Quadruplet-G | 0.865(0.01) | 0.055(0.02) | 0.423(0.01) | 0.228(0.01) | 0.675(0.04) | 0.051(0.01) |

*Table 11.* **Weakly supervised High-level partition results on *C. elegans Large* under different partition levels.** K-QD and U-QD are quartet distances on the *known* and *unknown* quartets, respectively.

| Method | RF↓ | △%RF↑ | QD↓ | △%QD↑ | △%K-QD↑ | △%U-QD↑ |
|---|---|---|---|---|---|---|
| **Partition Level: 3** | | | | | | |
| CellTreeQM | **0.793(0.01)** | **0.132(0.00)** | **0.071(0.00)** | **0.875(0.00)** | **0.989(0.00)** | -0.132(0.02) |
| CellTreeQM-G | **0.798(0.01)** | **0.122(0.02)** | **0.070(0.00)** | **0.877(0.00)** | **0.992(0.00)** | -0.137(0.02) |
| Triplet | 0.806(0.01) | 0.114(0.01) | 0.244(0.00) | 0.571(0.00) | 0.616(0.00) | **0.173(0.01)** |
| Triplet-G | 0.830(0.02) | 0.091(0.01) | 0.243(0.01) | 0.573(0.01) | 0.620(0.01) | **0.158(0.01)** |
| Quadruplet | 0.861(0.01) | 0.051(0.02) | 0.290(0.00) | 0.489(0.01) | 0.554(0.01) | -0.094(0.00) |
| Quadruplet-G | 0.863(0.01) | 0.047(0.01) | 0.292(0.00) | 0.486(0.00) | 0.552(0.00) | -0.100(0.01) |
| **Partition Level: 2** | | | | | | |
| CellTreeQM | 0.850(0.01) | 0.072(0.02) | **0.124(0.01)** | 0.781(0.02) | **0.996(0.00)** | **0.227(0.06)** |
| CellTreeQM-G | **0.839(0.00)** | 0.070(0.00) | **0.115(0.01)** | 0.797(0.02) | **0.996(0.00)** | **0.280(0.07)** |
| Triplet | **0.820(0.01)** | 0.098(0.01) | 0.239(0.01) | 0.576(0.01) | 0.760(0.01) | 0.099(0.01) |
| Triplet-G | 0.859(0.01) | 0.055(0.01) | 0.237(0.00) | 0.581(0.00) | 0.761(0.00) | 0.113(0.01) |
| Quadruplet | 0.917(0.00) | -0.010(0.01) | 0.235(0.00) | 0.586(0.00) | 0.864(0.00) | -0.137(0.01) |
| Quadruplet-G | 0.926(0.01) | -0.018(0.01) | 0.234(0.00) | 0.587(0.01) | 0.868(0.00) | -0.141(0.02) |
| **Partition Level: 1** | | | | | | |
| CellTreeQM | 0.902(0.00) | 0.006(0.01) | **0.261(0.01)** | **0.539(0.02)** | **0.999(0.00)** | **0.414(0.03)** |
| CellTreeQM-G | 0.904(0.00) | 0.014(0.01) | **0.298(0.00)** | **0.474(0.01)** | **0.999(0.00)** | **0.331(0.01)** |
| Triplet | **0.850(0.01)** | 0.063(0.01) | 0.364(0.00) | 0.357(0.00) | 0.831(0.01) | 0.228(0.00) |
| Triplet-G | **0.869(0.00)** | 0.035(0.01) | 0.369(0.00) | 0.350(0.01) | 0.821(0.02) | 0.222(0.01) |
| Quadruplet | 0.896(0.01) | 0.015(0.03) | 0.437(0.00) | 0.229(0.00) | 0.618(0.03) | 0.123(0.01) |
| Quadruplet-G | 0.885(0.01) | 0.022(0.02) | 0.433(0.01) | 0.235(0.01) | 0.637(0.02) | 0.127(0.01) |

*Table 12.* **Weakly supervised High-level partition results on *C. elegans Large* under different partition levels.** K-QD and U-QD are quartet distances on the *known* and *unknown* quartets, respectively. he reported values are means across three runs, with standard deviations in parentheses.

| Method | RF↓ | Δ%RF↑ | QD↓ | Δ%QD↑ | Δ%K-QD↑ | Δ%U-QD↑ |
|---|---|---|---|---|---|---|
| **Partition Level: 3** | | | | | | |
| CellTreeQM | 0.884 | 0.055 | **0.066** | **0.887** | **0.992** | 0.744 |
| CellTreeQM-G | 0.884 | 0.051 | **0.067** | **0.884** | **0.987** | **0.748** |
| Triplet | **0.846** | **0.105** | 0.272 | 0.531 | 0.573 | 0.576 |
| Triplet-G | **0.853** | **0.095** | 0.275 | 0.524 | 0.565 | 0.564 |
| Quadruplet | 0.887 | 0.062 | 0.312 | 0.461 | 0.519 | **0.746** |
| Quadruplet-G | 0.887 | 0.044 | 0.316 | 0.454 | 0.511 | 0.745 |
| **Partition Level: 2** | | | | | | |
| CellTreeQM | 0.918 | 0.025 | **0.117** | **0.798** | **0.992** | **0.646** |
| CellTreeQM-G | 0.921 | 0.022 | **0.124** | **0.786** | **0.997** | **0.620** |
| Triplet | **0.860** | **0.087** | 0.259 | 0.553 | 0.684 | 0.364 |
| Triplet-G | **0.856** | **0.088** | 0.263 | 0.547 | 0.680 | 0.357 |
| Quadruplet | 0.928 | 0.022 | 0.242 | 0.583 | 0.854 | 0.400 |
| Quadruplet-G | 0.918 | 0.018 | 0.239 | 0.589 | 0.862 | 0.412 |
| **Partition Level: 1** | | | | | | |
| CellTreeQM | **0.901** | 0.037 | **0.379** | **0.345** | **0.999** | **0.127** |
| CellTreeQM-G | 0.908 | 0.026 | **0.363** | **0.372** | **0.999** | **0.164** |
| Triplet | 0.908 | 0.050 | 0.390 | 0.327 | 0.890 | **0.140** |
| Triplet-G | **0.897** | 0.040 | 0.397 | 0.314 | **0.857** | 0.133 |
| Quadruplet | 0.918 | 0.025 | 0.465 | 0.200 | 0.641 | 0.053 |
| Quadruplet-G | 0.932 | 0.007 | 0.472 | 0.184 | 0.614 | 0.041 |

*Table 13.* Known, Partial, and Unknown counts and proportions for different known fractions in C. elegans datasets.

| | C. elegans Small | | C. elegans Mid | |
|---|---|---|---|---|
| **Known Fraction** | **Count** | **Prop.** | **Count** | **Prop.** |
| **Known Fraction: 0.8** | | | | |
| Known | 1,663,740 | 0.392 | 18,163,860 | 0.402 |
| Partial | 2,579,850 | 0.607 | 26,982,990 | 0.597 |
| Unknown | 5,985 | 0.001 | 66,045 | 0.001 |
| **Known Fraction: 0.5** | | | | |
| Known | 249,900 | 0.059 | 2,672,670 | 0.059 |
| Partial | 3,749,775 | 0.882 | 39,746,070 | 0.879 |
| Unknown | 249,900 | 0.059 | 2,794,155 | 0.062 |
| **Known Fraction: 0.3** | | | | |
| Known | 27,405 | 0.006 | 316,251 | 0.007 |
| Partial | 3,193,380 | 0.752 | 33,887,268 | 0.750 |
| Unknown | 1,028,790 | 0.242 | 11,009,376 | 0.244 |

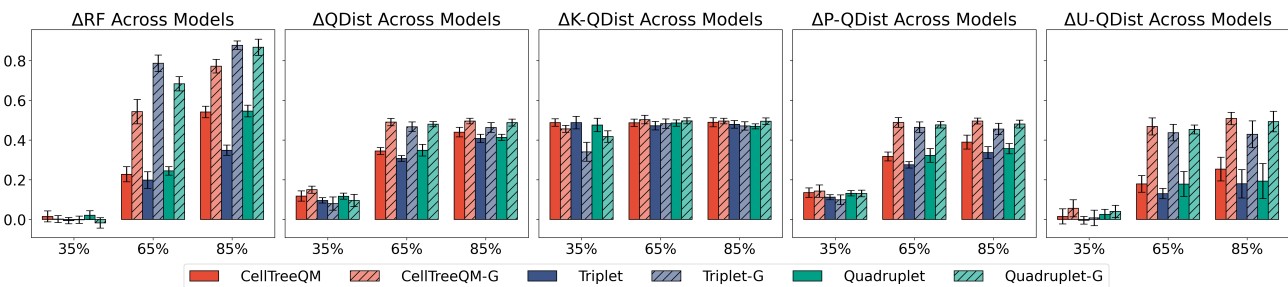

*Figure 19.* Results for **partially leaf-labeled** setting on simulation with 128 leaves dataset. x-axis is the percentage of known leaves.
.

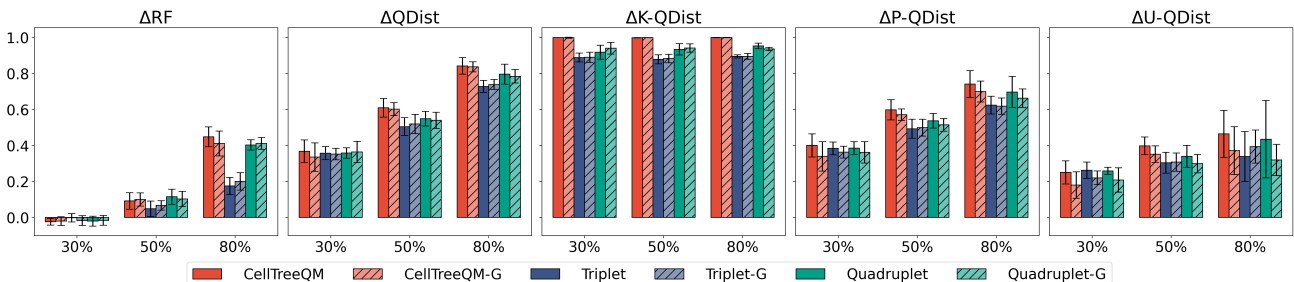

*Figure 20.* Results for **partially leaf-labeled** setting on *C. elegans* Small dataset. x-axis is the percentage of known leaves.

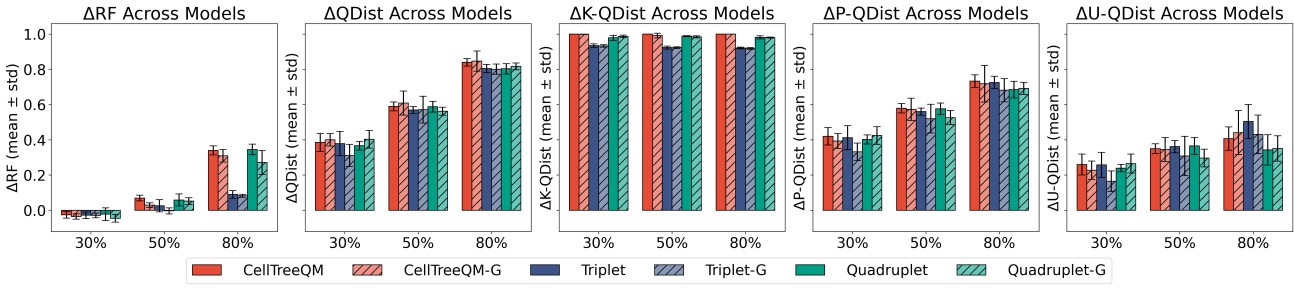

*Figure 21.* Results for **partially leaf-labeled** setting on *C. elegans* Mid dataset. x-axis is the percentage of known leaves.

*Table 14.* Weakly supervised Partial-labeled setting results across known fractions for C. elegans Small. The reported values are means across ten runs, with standard deviations in parentheses.

| Method | Train RF↓ | Δ%RF↑ | Δ%QD↑ | Δ%K-QD↑ | Δ%P-QD↑ | Δ%U-QD↑ |
|---|---|---|---|---|---|---|
| **Known Fraction: 0.8** | | | | | | |
| CellTreeQM | **0.024(0.03)** | **0.448(0.06)** | **0.842(0.05)** | **0.999(0.00)** | **0.742(0.07)** | **0.465(0.13)** |
| CellTreeQM-G | **0.031(0.04)** | 0.411(0.07) | **0.837(0.03)** | **0.999(0.00)** | **0.700(0.06)** | **0.371(0.13)** |
| Triplet | 0.454(0.06) | 0.175(0.05) | 0.728(0.03) | 0.895(0.01) | **0.624(0.05)** | 0.339(0.14) |
| Triplet-G | 0.419(0.07) | 0.201(0.05) | 0.739(0.03) | 0.895(0.02) | 0.618(0.05) | 0.394(0.09) |
| Quadruplet | 0.066(0.04) | **0.403(0.03)** | 0.796(0.06) | 0.953(0.02) | 0.697(0.09) | **0.435(0.22)** |
| Quadruplet-G | 0.060(0.04) | **0.411(0.03)** | 0.784(0.04) | 0.936(0.01) | 0.662(0.05) | 0.320(0.09) |
| **Known Fraction: 0.5** | | | | | | |
| CellTreeQM | **0.012(0.01)** | 0.092(0.05) | **0.609(0.05)** | **0.999(0.00)** | **0.598(0.06)** | **0.398(0.05)** |
| CellTreeQM-G | **0.016(0.01)** | 0.100(0.04) | **0.602(0.04)** | **0.999(0.00)** | **0.571(0.03)** | 0.352(0.05) |
| Triplet | 0.303(0.09) | 0.049(0.04) | 0.505(0.05) | 0.879(0.02) | 0.493(0.05) | **0.304(0.06)** |
| Triplet-G | 0.319(0.09) | 0.067(0.03) | 0.520(0.05) | 0.883(0.02) | 0.499(0.05) | 0.308(0.05) |
| Quadruplet | **0.023(0.02)** | 0.115(0.04) | 0.549(0.04) | 0.934(0.03) | **0.537(0.04)** | **0.340(0.06)** |
| Quadruplet-G | 0.039(0.03) | 0.103(0.04) | 0.540(0.05) | 0.942(0.02) | 0.515(0.04) | 0.299(0.05) |
| **Known Fraction: 0.3** | | | | | | |
| CellTreeQM | **0.000(0.00)** | -0.023(0.02) | **0.368(0.06)** | **1.000(0.00)** | **0.401(0.06)** | **0.250(0.06)** |
| CellTreeQM-G | **0.008(0.02)** | -0.020(0.02) | **0.336(0.08)** | **0.999(0.00)** | 0.340(0.08) | 0.180(0.07) |
| Triplet | 0.156(0.06) | -0.001(0.02) | 0.358(0.04) | 0.889(0.02) | 0.384(0.04) | **0.263(0.05)** |
| Triplet-G | 0.169(0.09) | -0.017(0.03) | 0.352(0.03) | **0.890(0.03)** | 0.363(0.03) | 0.220(0.04) |
| Quadruplet | 0.008(0.02) | -0.020(0.03) | 0.358(0.03) | 0.918(0.04) | **0.386(0.03)** | **0.259(0.02)** |
| Quadruplet-G | 0.029(0.05) | -0.016(0.03) | **0.364(0.06)** | 0.940(0.03) | **0.362(0.06)** | 0.208(0.07) |

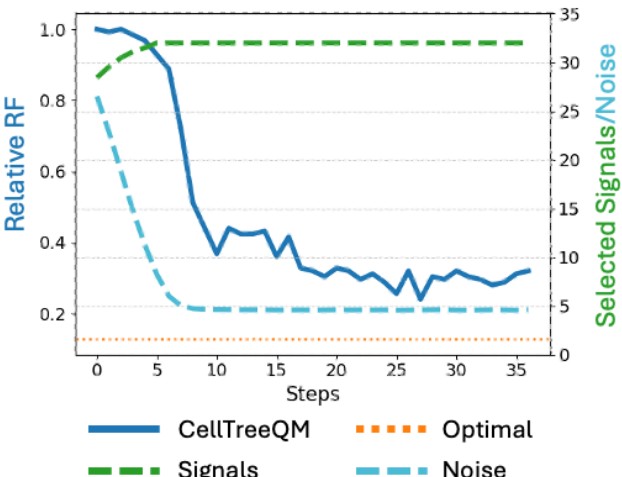

*Figure 22.* **Training dynamics of CellTreeQM in a purely unsupervised setting on a simulated dataset.** The left axis reports the relative RF of the reconstructed tree (blue) compared to the optimal RF (orange dotted line), while the right axis indicates the number of selected signal (green dashed) and noise (blue dashed) features over training steps. Early on, more noise features are included, resulting in poorer performance; as training proceeds, CellTreeQM discards the noise and approaches the optimal reconstruction.

*Table 15.* Weakly supervised Partial-labeled setting results across known fractions for C. elegans Mid. The reported values are means across five runs, with standard deviations in parentheses

| Method | Train RF↓ | △%RF↑ | △%QD↑ | △%K-QD↑ | △%P-QD↑ | △%U-QD↑ |
|---|---|---|---|---|---|---|
| **Known Fraction: 0.8** | | | | | | |
| CellTreeQM | **0.108(0.02)** | **0.339(0.03)** | **0.840(0.02)** | **1.000(0.00)** | **0.733(0.04)** | 0.407(0.07) |
| CellTreeQM-G | **0.150(0.05)** | 0.309(0.04) | **0.847(0.06)** | **1.000(0.00)** | 0.718(0.10) | **0.441(0.13)** |
| Triplet | 0.628(0.02) | 0.090(0.02) | 0.804(0.02) | 0.921(0.01) | **0.725(0.04)** | **0.503(0.10)** |
| Triplet-G | 0.656(0.02) | 0.082(0.01) | 0.801(0.03) | 0.918(0.01) | 0.681(0.07) | 0.431(0.11) |
| Quadruplet | 0.169(0.02) | **0.345(0.03)** | 0.803(0.03) | 0.981(0.01) | 0.684(0.05) | 0.343(0.09) |
| Quadruplet-G | 0.250(0.07) | 0.271(0.07) | 0.817(0.02) | 0.980(0.00) | 0.691(0.03) | 0.351(0.07) |
| **Known Fraction: 0.5** | | | | | | |
| CellTreeQM | **0.055(0.01)** | 0.069(0.02) | **0.589(0.03)** | **1.000(0.00)** | **0.579(0.03)** | 0.349(0.03) |
| CellTreeQM-G | 0.180(0.10) | 0.029(0.01) | **0.609(0.07)** | **0.991(0.01)** | 0.573(0.06) | 0.344(0.07) |
| Triplet | 0.491(0.03) | 0.025(0.03) | 0.569(0.02) | 0.923(0.01) | 0.560(0.02) | **0.361(0.04)** |
| Triplet-G | 0.509(0.06) | -0.004(0.02) | 0.571(0.08) | 0.923(0.01) | 0.520(0.08) | 0.308(0.11) |
| Quadruplet | **0.093(0.02)** | 0.058(0.03) | 0.587(0.03) | 0.988(0.00) | **0.576(0.03)** | **0.365(0.05)** |
| Quadruplet-G | 0.180(0.10) | 0.052(0.02) | 0.562(0.02) | 0.984(0.01) | 0.526(0.04) | 0.296(0.05) |
| **Known Fraction: 0.3** | | | | | | |
| CellTreeQM | **0.000(0.00)** | -0.026(0.02) | 0.385(0.05) | **1.000(0.00)** | **0.420(0.05)** | **0.259(0.06)** |
| CellTreeQM-G | **0.027(0.03)** | -0.036(0.02) | **0.400(0.04)** | **0.999(0.00)** | 0.393(0.04) | 0.227(0.05) |
| Triplet | 0.275(0.10) | -0.028(0.02) | 0.378(0.07) | 0.935(0.01) | 0.412(0.07) | 0.257(0.07) |
| Triplet-G | 0.306(0.13) | -0.029(0.01) | 0.311(0.06) | 0.932(0.01) | 0.332(0.05) | 0.164(0.06) |
| Quadruplet | 0.008(0.02) | -0.022(0.04) | 0.366(0.02) | 0.979(0.01) | 0.402(0.03) | 0.239(0.02) |
| Quadruplet-G | 0.067(0.06) | -0.045(0.02) | **0.403(0.05)** | 0.987(0.01) | **0.424(0.05)** | **0.265(0.05)** |

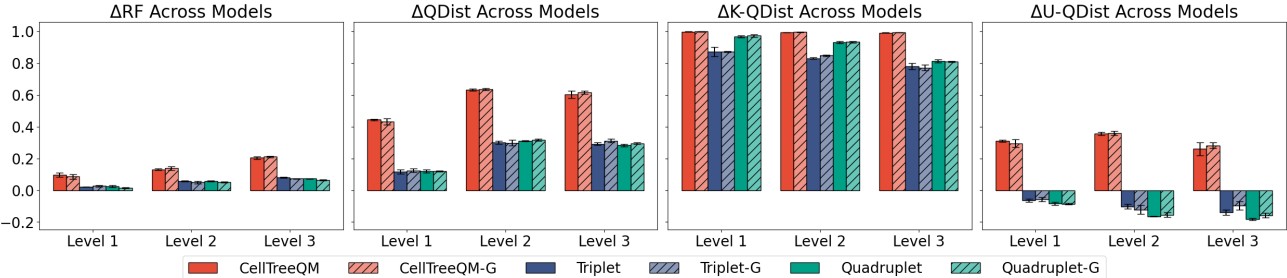

*Figure 23.* Results for **high-level partitioning** setting on CRISPR dataset 3435_NT_T1. x-axis is the level of partition.

*Table 16.* **Weakly supervised high-level partitioning setting results on CRISPR dataset 3435_NT_T1 under different partition levels.** K-QD and U-QD are quartet distances on the *known* and *unknown* quartets, respectively. The reported values are means across three runs, with standard deviations in parentheses.

| Method | RF↓ | △%RF↑ | QD↓ | △%QD↑ | △%K-QD↑ | △%U-QD↑ |
|---|---|---|---|---|---|---|
| **Partition Level: 3** | | | | | | |
| CellTreeQM | **0.791(0.01)** | **0.204(0.01)** | **0.199(0.01)** | **0.603(0.02)** | **0.992(0.00)** | **0.261(0.04)** |
| CellTreeQM-G | **0.784(0.01)** | **0.211(0.01)** | **0.193(0.00)** | **0.616(0.01)** | **0.994(0.00)** | **0.282(0.02)** |
| Triplet | 0.914(0.00) | 0.079(0.00) | 0.355(0.00) | 0.292(0.01) | 0.779(0.02) | -0.139(0.02) |
| Triplet-G | 0.921(0.00) | 0.073(0.00) | 0.345(0.01) | 0.311(0.01) | 0.771(0.02) | -0.096(0.03) |
| Quadruplet | 0.921(0.00) | 0.073(0.00) | 0.359(0.00) | 0.284(0.01) | 0.812(0.01) | -0.182(0.01) |
| Quadruplet-G | 0.928(0.00) | 0.066(0.00) | 0.353(0.00) | 0.296(0.01) | 0.810(0.00) | -0.158(0.01) |
| **Partition Level: 2** | | | | | | |
| CellTreeQM | **0.863(0.01)** | **0.132(0.01)** | **0.184(0.00)** | **0.632(0.01)** | **0.994(0.00)** | **0.356(0.01)** |
| CellTreeQM-G | **0.856(0.01)** | **0.138(0.01)** | **0.182(0.00)** | **0.636(0.01)** | **0.996(0.00)** | **0.361(0.01)** |
| Triplet | 0.937(0.00) | 0.057(0.00) | 0.350(0.01) | 0.302(0.01) | 0.831(0.00) | -0.103(0.01) |
| Triplet-G | 0.944(0.01) | 0.050(0.01) | 0.351(0.01) | 0.299(0.02) | 0.848(0.00) | -0.121(0.03) |
| Quadruplet | 0.935(0.00) | 0.059(0.00) | 0.346(0.00) | 0.310(0.00) | 0.931(0.01) | -0.165(0.00) |
| Quadruplet-G | 0.944(0.00) | 0.050(0.00) | 0.342(0.00) | 0.317(0.01) | 0.934(0.00) | -0.155(0.01) |
| **Partition Level: 1** | | | | | | |
| CellTreeQM | **0.896(0.01)** | **0.098(0.01)** | **0.279(0.00)** | **0.444(0.00)** | **0.999(0.00)** | **0.310(0.01)** |
| CellTreeQM-G | **0.908(0.01)** | **0.086(0.01)** | **0.285(0.01)** | **0.432(0.02)** | **0.999(0.00)** | **0.295(0.03)** |
| Triplet | 0.973(0.00) | 0.020(0.00) | 0.443(0.01) | 0.117(0.01) | 0.872(0.03) | -0.065(0.01) |
| Triplet-G | 0.966(0.01) | 0.027(0.01) | 0.439(0.01) | 0.124(0.01) | 0.873(0.00) | -0.057(0.01) |
| Quadruplet | 0.968(0.01) | 0.025(0.01) | 0.441(0.00) | 0.121(0.01) | 0.968(0.01) | -0.084(0.01) |
| Quadruplet-G | 0.977(0.00) | 0.016(0.00) | 0.441(0.00) | 0.120(0.00) | 0.973(0.01) | -0.086(0.00) |

*Table 17.* **Weakly supervised high-level partitioning setting results on CRISPR dataset 3435_NT_T6 under different partition levels.** K-QD and U-QD are quartet distances on the *known* and *unknown* quartets, respectively. The reported values are means across three runs, with standard deviations in parentheses.

| Method | RF↓ | △%RF↑ | QD↓ | △%QD↑ | △%K-QD↑ | △%U-QD↑ |
|---|---|---|---|---|---|---|
| **Partition Level: 3** | | | | | | |
| CellTreeQM | **0.705(0.01)** | **0.287(0.01)** | **0.151(0.02)** | **0.740(0.03)** | **0.984(0.01)** | **0.236(0.11)** |
| CellTreeQM-G | **0.682(0.01)** | **0.310(0.01)** | **0.124(0.01)** | **0.787(0.02)** | **0.989(0.00)** | **0.369(0.06)** |
| Triplet | 0.780(0.01) | 0.211(0.01) | 0.281(0.01) | 0.517(0.02) | 0.730(0.03) | 0.077(0.03) |
| Triplet-G | 0.780(0.01) | 0.211(0.01) | 0.280(0.01) | 0.520(0.01) | 0.734(0.01) | 0.075(0.02) |
| Quadruplet | 0.852(0.01) | 0.138(0.01) | 0.318(0.01) | 0.454(0.02) | 0.748(0.01) | -0.155(0.03) |
| Quadruplet-G | 0.867(0.01) | 0.123(0.01) | 0.317(0.00) | 0.456(0.00) | 0.753(0.01) | -0.159(0.00) |
| **Partition Level: 2** | | | | | | |
| CellTreeQM | **0.780(0.01)** | **0.211(0.01)** | **0.227(0.01)** | **0.611(0.02)** | **0.993(0.00)** | **0.253(0.03)** |
| CellTreeQM-G | **0.792(0.01)** | **0.199(0.01)** | **0.220(0.00)** | **0.623(0.01)** | **0.991(0.00)** | **0.276(0.02)** |
| Triplet | 0.837(0.01) | 0.153(0.01) | 0.379(0.01) | 0.349(0.01) | 0.673(0.02) | 0.046(0.02) |
| Triplet-G | 0.852(0.00) | 0.138(0.00) | 0.378(0.01) | 0.352(0.02) | 0.694(0.02) | 0.030(0.02) |
| Quadruplet | 0.898(0.01) | 0.092(0.01) | 0.397(0.01) | 0.319(0.01) | 0.820(0.02) | -0.151(0.02) |
| Quadruplet-G | 0.902(0.01) | 0.088(0.01) | 0.379(0.01) | 0.350(0.01) | 0.858(0.01) | -0.127(0.02) |
| **Partition Level: 1** | | | | | | |
| CellTreeQM | **0.845(0.01)** | **0.146(0.01)** | **0.291(0.01)** | **0.501(0.01)** | **1.000(0.00)** | **0.429(0.01)** |
| CellTreeQM-G | **0.962(0.01)** | 0.027(0.01) | 0.542(0.03) | 0.070(0.05) | **0.998(0.00)** | -0.063(0.06) |
| Triplet | 0.947(0.01) | 0.042(0.01) | 0.521(0.01) | 0.106(0.02) | 0.862(0.07) | -0.003(0.01) |
| Triplet-G | 0.958(0.01) | 0.031(0.01) | 0.534(0.01) | 0.084(0.02) | 0.816(0.02) | -0.020(0.02) |
| Quadruplet | 0.977(0.00) | 0.011(0.00) | 0.604(0.03) | -0.036(0.06) | 0.142(0.22) | -0.061(0.03) |
| Quadruplet-G | 0.970(0.01) | 0.019(0.01) | 0.613(0.01) | -0.052(0.02) | 0.299(0.06) | -0.103(0.02) |

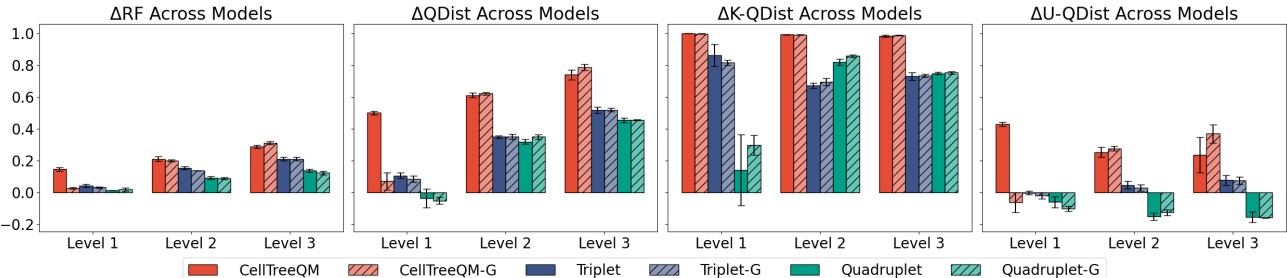

*Figure 24.* Results for **high-level partitioning** setting on CRISPR dataset 3435_NT_T6. x-axis is the level of partition.
.

