# OpenReview forum: "Reconstructing Cell Lineage Trees from Phenotypic Features with Metric Learning"
_ICML.cc/2025/Conference — ICML 2025 poster_

### Official Review · Reviewer_FYFt · 2025-03-02

**Overall Recommendation:** 4

**Summary:**

This work addresses an important problem: reconstructing cell lineage trees, which provide insights of how diverse cell types arise from a single progenitor. More to the point, the authors set an additional challenge, which is to use transcriptomic data to construct cell linage trees, which sets them apart from the classical lineage inference methods that rely on genetic barcoding, CRISPR-based lineage tracing, or phylogenetic distance methods. In other words, they attempt to answer the question  whether high-content molecular phenotypes give enough information for tree reconstruction and in doing  so they define a novel method that can be used to build cell lineage trees using such data.

The crux of the proposed method is to rely on the traditional framework of distance-based lineage reconstruction, which is a known NP-hard problem for which many approximation algorithms exist, such as the neighbor-joining algorithm, which they also use in their work. The novelty, however, consists in shifting the problem to a metric-learning one, whereby the idea is to learn an embedding function that maps data points to a space where distances approximate an additive metric. By doing so, the intuition is that it will facilitate the task of tree construction, if quartet constraints -- which are crucial for accurate lineage inference -- are well learned or provided. In an ideal scenario where all tree quartets are known, the approach should work near-perfectly. However, in more realistic settings where many quartets must be inferred, errors can propagate, and the final tree may only be an approximation of the true lineage structure.

In summary, the metric learning problem relies on the definition of two loss terms that strive at verifying the four-point condition (or quartet constraint). In addition, the authors propose a regularization term to prevent the learned embedding to drift too much away from the original data, and a gating mechanism (a sort of feature selection mechanism) that emphasize only lineage-relevant features of the high-dimensional transcriptomic data they use.

Experiments are mainly devoted to verify the benefits of the proposed embedding method, rather than comparing the overall approach to state of the art. Then, they compare to traditional metric learning methods inspired by contrastive learning losses. Since CellTreeQM also uses a metric-learning approach, comparing against standard loss functions provides a fair assessment of whether their tree-based additivity loss truly improves tree reconstruction.

## POST REBUTTAL
Thank you for the rebuttal and the discussions. I maintain my positive score for this paper.

**Claims And Evidence:**

I think that most claims are partially supported by evidence in this work.

* Claim 1: inferring or reconstructing the cell lineage tree from features measured on the individual cells is an important challenge that they address in this work. This is the essence of the proposed method, that uses transcriptomic data.

* Claim 2: the open question the authors address is whether high-content molecular phenotypes give enough information for tree reconstruction. The experiments provide a positive answer to this question.

* Claim 3: Cell-TreeQM efficiently reconstructs lineage structures under weak supervision and limited data, providing a scalable framework for uncovering cell lineage relationships. This is supported by the experiments in section 7.

**Essential References Not Discussed:**

N/A

**Experimental Designs Or Analyses:**

Yes, I checked experimental design and analyses and they are appropriate in my opinion. As mentioned above, the authors mainly focus on metric learning, and compare their proposed learned embeddings to that learned with alternative contrastive losses, showing that the key idea of enforcing the quartet constraints is indeed important. They do so in a variety of settings, for both synthetic and real data.

**Methods And Evaluation Criteria:**

Let me separate methods from evaluation.

On the methodological side, the proposed idea to project high-dimensional data into a latent representation that is amenable to preserve an appropriate distance relationship between data points is appropriate, relevant and novel, to the best of my knowledge. The proposed methodology is clearly exposed, except for a detail.
The overall narrative of section 4.1 is to perform a joint optimization of tree construction and metric learning, as defined in the last optimization expression at the end of the section. However, the proposed method is *not* a joint optimization: first, the authors propose a method to learn an embedding, and subsequently, this embedding is used within the distance-based lineage reconstruction formulation and fed to the well-known neighbor-joining algorithm.
The authors take steps to ensure that the learned distances conform to a tree metric, which helps mitigate the issue of separating the embedding and tree reconstruction steps. However, it still does not fully jointly optimize the tree and the embedding function together. Instead, it shapes the embedding space to be more tree-like before applying a separate tree reconstruction algorithm. This is a reasonable approximation, but not a true joint optimization approach, which would require incorporating the actual tree inference step directly into the learning process.

On the experimental side, the authors do a good job in evaluating the proposed embedding against that learned through traditional contrastive losses. Although the selected metric learning baselines make sense for their approach, a more complete experimental evaluation could have included comparisons to state-of-the-art trajectory inference methods , which would help show how well CellTreeQM recovers lineage structure compared to methods that infer differentiation trajectories. Since the paper already discusses trajectory inference in Appendix C, a quantitative comparison would strengthen their claims. Moreover, instead of just using the neighbor-joining algorithm, they could have tested alternative distance-based tree reconstruction methods, which could have revealed whether their learned embeddings work well across different methods.
Finally, if a dataset existed which had both phenotypic features and genetic barcodes, they could compare their phenotypic-based tree to a gold-standard lineage tree inferred from CRISPR mutations, which could have shown whether their learned embeddings truly capture lineage relationships or if they primarily reflect phenotypic similarity.

**Other Comments Or Suggestions:**

N/A

**Other Strengths And Weaknesses:**

This is a good work, the only comments I outlined above are:

* Method: pay attention to the claim of a joint optimization problem, versus breaking the problem in two stages.

* Experiments: while it is valid to compare the proposed method to metric learning baselines, I think this work would be much more valuable to the biology community if the experimental section was extended to cover 1) alternative tree construction algorithms (not only neighbor-joining), and 2) more end-to-end comparisons, although this is hindered by the probable absence of relevant datasets that could be used both by comparing phenotypic-based tree to a gold-standard lineage tree inferred from CRISPR mutations.

**Questions For Authors:**

* Can you please provide clarifications about the remark of a joint optimization problem vs. breaking the problem into first learning an appropriate embedding, and then applying a well-known tree construction algorithm? Did I misinterpret your approach? If not, can you maybe add some comments about the sub-optimality of the resulting lineage reconstruction when compared to a truly joint optimization approach?

* Would it be possible to add experiments that use an alternative tree construction algorithm than the neighbor-joining one? You may just keep everything you've learned through the embedding, and apply another technique to further confirm the versatility of the learned embeddings.

**Relation To Broader Scientific Literature:**

This is a multidisciplinary project, lying at the intersection of machine learning and biology. I think the authors did a very good job in the main paper, and the interested reader can find much more in the appendix, especially Appendix A (which is an short introduction to the context of their study) and in Appendix C.

**Theoretical Claims:**

N/A

---

> ### Author Rebuttal · Authors · 2025-04-01
>
> ## Concern 1: Joint optimization claim
> ### Response
> We thank the reviewer for recognizing the importance of the cell lineage reconstruction problem and for raising this valuable point about our optimization formulation. We agree that Section 4.1 could be clearer in distinguishing between joint and staged optimization strategies, and we appreciate the opportunity to clarify our intent.
>
> As the reviewer correctly notes, CellTreeQM adopts a two-stage framework:
>
> 1. Learn an embedding space using quartet-based constraints that promotes tree-like geometry;
> 2. Apply a standard phylogenetic algorithm (e.g., Neighbor-Joining) to reconstruct the tree from the learned pairwise distances.
>
> Although staged in practice, we emphasize that **most of the tree inference is effectively encoded in the embedding step**. Once the embedding space approximates tree-additive distances, standard reconstruction methods reliably recover the topology (as shown in Appendix A.2, line 742). Figure 4 further illustrates that both Train RF and Test RF steadily decrease during training, indicating that the learned representation is progressively converging toward a tree-consistent structure. This is conceptually aligned with heuristic quartet-based methods (e.g. Quartet Puzzling), which also tackle the NP-hard tree reconstruction problem by optimizing over informative subsets of constraints.
>
> To better reflect our approach, we updated the manuscript to clearify that we revise the following object function
> $$\min_{f, T} \|D(f(x)) - D_T\|_2^2 + \lambda \Omega(f),$$
>
> into quartet-based formulation
>
> $$\min_{f}\sum_{q \in \mathcal{Q}}\mathcal L( D\bigl(f(x_q)\bigr), D_{T0}\bigl(x_q\bigr))+\lambda \Omega(f),$$
> where $Q$ is the set of known quartets and $D_{T0}$ is the pairwise distance for the known quartets.
>
> When $T$ is known, this objective reduces to a supervised metric learning problem. However, as we highlight in the manuscript, full lineage trees are often unavailable in biological datasets. Instead, researchers may have access to partial lineage information—such as clade-level groupings—while finer substructures remain uncertain. In these settings, we found quartet constraints offer a flexible and localized way to encode supervision and capture tree-additive properties.
>
> ## Concern 2: comparison between trajectory inference and cell lineage reconstruction
> ### Response
> As discussed in our response to Reviewer ekW4’s Concern 1, trajectory inference and lineage reconstruction address fundamentally different problems. To further illustrate this distinction, we conducted additional experiments using the C. elegans Small dataset.
>
> Using Monocle3, we constructed a principal graph from unsupervised clusters and colored cells by lineage annotation ([figure](https://anonymous.4open.science/api/repo/celltreeqm-rebuttal-7D65/file/monocle.png?v=b00073b0)). The resulting graph shows that many terminal lineages are incorrectly represented as internal nodes, highlighting a mismatch with known lineage structure.
>
> Because Monocle3 cannot incorporate known labels into graph construction, we also computed lineage centroids and built a minimum spanning tree (MST) to model their relationships ([HotSpot](https://anonymous.4open.science/api/repo/celltreeqm-rebuttal-7D65/file/mst.png?v=431c273e), [ASCII](https://anonymous.4open.science/api/repo/celltreeqm-rebuttal-7D65/file/mst_ascii.png?v=f4902aed)). Again, several terminal lineages are placed as internal nodes. HotSpot only shows the densely connect dots and ASCII shows the full MST.
>
> These results confirm that standard trajectory inference methods do not accurately reflect true lineage trees and are also not directly comparable to lineage reconstruction.
>
> ## Concern 3: Limited tree reconstruction methods
> ### Response
> We appreciate the suggestion to evaluate additional reconstruction algorithms beyond Neighbor-Joining (NJ). We assessed four alternatives—UPGMA, FastME, Ward, and Single linkage on the C. elegans Small dataset ([figure](https://anonymous.4open.science/api/repo/celltreeqm-rebuttal-7D65/file/reconstruct_method.png?v=d604180e)).
>
> Across all methods, CellTreeQM consistently achieves lower RF distances on both train and test sets, consistently demonstrating strong learning and generalization ability. The radar plot shows relative RF improvement across all five reconstruction algorithms comparing with Triplet and Quadruplet.
>
> Interestingly, Ward’s method slightly outperforms NJ, despite NJ’s theoretical alignment with additive trees. This suggests our embeddings may better support variance-based clustering in certain settings. We plan to investigate further in future work.
>
> ## Concern 4: Lack of gold-standard lineage tree comparison
> ### Response
> We thank the reviewer for this important point. Please see our response of vktx Concern3.

---

> > ### Comment · Reviewer_FYFt · 2025-04-02
> >
> > Thank you for the rebuttal and the clarifications. I keep my positive score.

---

### Official Review · Reviewer_ekW4 · 2025-03-04

**Overall Recommendation:** 2

**Summary:**

This paper introduces **CellTreeQM**, a novel deep learning-based method for reconstructing cell lineage trees using single-cell RNA sequencing data.
The method leverages a **Transformer architecture** combined with **metric learning** to optimize the geometry of an embedding space, enabling accurate reconstruction of lineage trees even with limited supervision or noisy data.
Unlike traditional approaches relying on CRISPR barcoding or heuristic distance-based methods, CellTreeQM explicitly formulates lineage reconstruction as a metric learning problem and incorporates **triplet and quadruplet loss functions** to enforce tree-like relationships in the embedding space.
The authors also establish a benchmark for lineage reconstruction and demonstrate that CellTreeQM outperforms existing methods in terms of accuracy, robustness, and scalability.

**Claims And Evidence:**

Yes.

**Essential References Not Discussed:**

No.

**Experimental Designs Or Analyses:**

1. This paper's experimental section only discusses comparisons with Quadruplet and Triplet, but in reality, there are many state-of-the-art (SOTA) cell lineage methods that have not been compared.
2. Many trajectory inference methods for constructing cell lineage from single-cell expression data ultimately yield tree-shaped results. Could we perform a simple comparison with these methods, e.g monocle3, Slingshot?
3. The Transformer encoder module has not been ablated, and while Transformers are well-known for their effectiveness on large datasets, methods like VAE may be more effective on the current small dataset.

**Methods And Evaluation Criteria:**

Yes.

**Other Comments Or Suggestions:**

No.

**Other Strengths And Weaknesses:**

1. Why wasn't the effect of feature gates compared in the weakly supervised scenario?

**Questions For Authors:**

No.

**Relation To Broader Scientific Literature:**

1. **Advancements in Cell Lineage Reconstruction**: The paper builds upon existing methods, which uses CRISPR barcoding for lineage tracing, by introducing a deep learning-based approach that leverages **metric learning** and **Transformer architectures** to improve accuracy and robustness in lineage reconstruction.

2. **Integration with Metric Learning**: It extends the application of **metric learning** techniques, commonly used in classification and clustering tasks, to the domain of cell lineage reconstruction, enhancing the ability to capture tree-like relationships in high-dimensional data.

**Theoretical Claims:**

Yes. The Four-Point Condition ensures that distance matrix can be constructed into a tree, but whether it is applicable to the current task is unclear, as there has been no previous exploration in this regard.

---

> ### Author Rebuttal · Authors · 2025-04-01
>
> ## Concern 1: Lack of comparisons to SOTA methods
> > There are many state-of-the-art (SOTA) cell lineage methods that have not been compared.
> ### Response
> Thank you for raising the concern regarding comparisons to state-of-the-art (SOTA) cell lineage methods. Below, we clarify why existing methods are not directly comparable to our approach.
>
> **Phenotype-Based vs. Genotype-Based Cell Lineage Methods**
>
> In the second paragraph of the Introduction and Section 3, we review computational methods for cell lineage reconstruction, which primarily rely on genotype information. In contrast, we focus on phenotype-based lineage reconstruction—a fundamentally different and much more challenging problem setting. To our knowledge, and as noted by Reviewers RR5y and FYFt, CellTreeQM is the first work to formulate and solve this specific task.
>
> We propose that both observed empirical data and biological process considerations suggest that the standard full feature space (i.e., the transcriptome) does not directly contain cell lineage information. Current time-trajectory algorithms are not focused on recovering the historical cell lineage relationships. Rather, the algorithms try to find a piecewise one-dimensional parameterization of cells in the transcriptome feature space as a model of “phenotype dynamics” solely based on trivial metric (such as l1 and l2 distance); i.e., how the phenotypes of the cells might dynamically evolve through cell system processes without a reference to underlying cell lineage processes. In fact, phenotype dynamics might occur without any cell division.
>
> **Baseline Selection**
>
> We chose triplet and quadruplet contrastive losses as baselines because these are widely used in related contrastive learning tasks. As we show, CellTreeQM’s tailored assumption about the structure of cell lineage data leads to superior performance compared to these generic approaches.
>
> **Clarifying the Role of Trajectory Inference**
>
> By “SOTA,” the reviewer may also be referring to trajectory inference methods. (Other methods that operate over sequence data are not applicable to the gene expression data studied in our paper.) While trajectory inference and lineage reconstruction are closely related, they address distinct goals. We provide a detailed comparison in Appendix C.1 (moved from the main text due to space constraints). Indeed, Reviewers RR5y and FYFt also noted the conceptual differences between these two tasks.
> - Methodological Differences
>
> Trajectory inference is typically an **unsupervised** pipeline comprising (1) data preprocessing, (2) dimensionality reduction, (3) cell clustering, and (4) learning a graph representation of cluster relationships. Such methods usually yield average progression trends rather than a cell-by-cell hierarchical lineage.
> - Mismatch in Assumptions
>
> Trajectory inference assumes the cells dynamically change within the full feature space and the goal is to trace this dynamics. The dynamics can be decoupled from cell lineage history. However, if the dataset has the cells at every stage of cell development, the algorithms could approximate the cell lineage histories, assuming the local changes in phenotypes are due to cell lineage development. However, as noted in various examples in the paper, even under this setting, the full feature space cell distributions typically do not reflect cell lineage history, requiring additional learning as we propose in our paper.
>
> ## Concern 2: No ablation of the Transformer encoder and compare to VAE
>
> ### Response
> We thank the reviewer for considering the model architecture of CellTreeQM. We would like to clarify that the motivation and empirical support for using a Transformer encoder are provided in Section 5.4. Specifically, we compared fully connected (FC) networks and Transformer encoders on both the C. elegans Small and Large datasets, and found that Transformers consistently outperformed FCs, as shown in Table 5.
>
> While we acknowledge the relevance of concurrent works such as TreeAVE, which focus on generative modeling with hierarchical latent variables, our goal differs. CellTreeQM does not aim to model the full generative process. Rather, we assume that lineage-related signals lie in a subspace of the observed features. Instead of relying on reconstruction losses typical in VAE-based models, we introduce the Deviation Loss, \Omega, to prevent the learned embedding from drifting too far from the original data.
>
> ## Concern 3: No results of feature gating for weakly supervised scenario
>
> ### Response
> We apologize for the confusion. Due to space constraints, we omitted the results of feature gating in the weakly supervised setting from the main text. However, these results are included in Tables 10–14 of the appendix. While feature gating provides marginal gains on real datasets, they demonstrate that the gating mechanism can still contribute positively, particularly in settings where the signal-to-noise ratio is low.

---

### Official Review · Reviewer_vktx · 2025-03-12

**Overall Recommendation:** 3

**Summary:**

The authors propose an algorithm CellTreeQM to reconstruct lineage relationships from phenotype data (unlike genotype data, which has been the main focus in the area of lineage reconstruction). The main new idea in the paper is the use of loss function based on four point condition which ensures that embedding eventually resembles an unrooted tree. At the same time this four point loss may create a distortion from allowing the model to fit to the original data; so the authors have an additional distortion loss term as well. This careful design of loss function is in my opinion the main new idea in the paper.

**Claims And Evidence:**

Yes

**Essential References Not Discussed:**

While the references on tree and random forest embedding are discussed in the literature cited; it would be appropriate to discuss the connection between them directly; as I can imagine a randomized tree embedding (Bartal et al be cited) to be used instead of just a tree embedding. The output would be a probability distribution over trees as opposed to a single tree.

**Experimental Designs Or Analyses:**

The experiments on C.elegans dataset are adequately explained in the supplement.

**Methods And Evaluation Criteria:**

Yes

**Other Comments Or Suggestions:**

See question about using random forests instead of trees above

**Other Strengths And Weaknesses:**

The paper uses reasonably deep theory background to solve an interesting problem -- lineage reconstruction. However, the authors have used just one dataset C.elegans for their experiments and do not prove any new theory results either. This makes me wonder if the contribution crosses the threshold for acceptance to ICML. Hence my rating.

**Questions For Authors:**

see above

**Relation To Broader Scientific Literature:**

Adequate (but also see comment below re metric embeddings)

**Theoretical Claims:**

While the authors discuss relevant theory background and how the tree reconstruction problem relates to existing theory, they do not appear to claim any significant new theoretical results (no new theorems/proofs).

---

> ### Author Rebuttal · Authors · 2025-03-31
>
> ## Concern 1: No theoretical results
> > they do not appear to claim any significant new theoretical results (no new theorems/proofs).
>
> ### Response
> While we do not present new theorems or proofs, our contribution lies in empirically demonstrating that the four-point condition serves as a strong prior for learning a generalizable embedding function—one that can correctly predict unseen quartet topologies. We hope we are demonstrating two new ideas: (1) with respect to learning tree-graphs, jointly learning the whole tree might be limited both algorithmically and by available supervised knowledge, but decomposition to quartet tree sub-trees allows utilization of partial information; (2) rather than using standard contrastive learning paradigm, explicitly utilizing a tree-graph metric theorem significantly improves the learning results.
>
> Following this comment, we have added additional test data (vktx Concern 3) and also additional phylogeny methods  (FYFt Concern 3) as well as other single cell trajectory methods (FYFt Concern 2) for comparison. We note that actual empirical data with validated annotated lineages are extremely rare, but we understand that greater degree of testing would be desirable.
>
> ## Concern 2: Missing literature
> > While the references on tree and random forest embedding are discussed in the literature cited; it would be appropriate to discuss the connection between them directly; as I can imagine a randomized tree embedding (Bartal et al be cited) to be used instead of just a tree embedding. The output would be a probability distribution over trees as opposed to a single tree.
>
> ### Response
> Thank you for this suggestion for the background literature that we were not aware of. Indeed, an observed empirical distance matrix could be considered a graph metric and bounded  by expectation over a distribution over a tree graph. This distribution might be mapped to a distribution of possible cell-lineage tree graphs in a similar framework as Bayesian phylogenetic algorithms. In our case, our assumption is that the original feature space and the resulting empirical metric does not reflect the lineage tree-graph, even as, say, a dominating metric. However, it might be possible to have a construction that computes a graph metric from the learned latent space and then associate it to a distribution over a tree metric ensemble. One might have to develop additional criteria for the tree metric such as “concentration” to incorporate into a learning scheme.
>
> While the full exposition as above is out of the scope of our current work, we have added the following references to our background literature:
>
> - Yair Bartal. Probabilistic approximations of metric spaces and its algorithmic
> applications. In Proceedings of the 37th IEEE Symposium on Foundations of
> Computer Science (FOCS), pages 184–193, 1996.
> - Yair Bartal. On approximating arbitrary metrics by tree metrics. In Proceedings of
> the 30th ACM Symposium on Theory of Computing (STOC), pages 161–168, 1998.
>
> ## Concern 3: Limited dataset
> > However, the authors have used just one dataset C.elegans for their experiments and do not prove any new theory results either. This makes me wonder if the contribution crosses the threshold for acceptance to ICML. Hence my rating.
>
> ### Response
> We thank the reviewer for this important point. In response, we have significantly expanded our experiments to include two additional reference datasets (including a CRISPR-based lineage tracing dataset from Yang et al., 2022), four more phylogenetic methods, and two trajectory inference methods.
>
> For the CRISPR dataset, we used cell lineages 3435_NT_T1 (151 cells) and 3435_NT_T6 (91 cells), both derived from mESC clone 1D5 with non-targeting sgRNA. Ground truth trees were reconstructed using Cassiopeia. Due to ambiguity in the tree structure, we sampled binary trees from the full lineage.
>
> We conducted weakly supervised learning with high-level partition prior in level 1, 2, and 3 with three repetitions. The detailed results are shown in [3435_NT_T1 BarPlot](https://anonymous.4open.science/r/celltreeqm-rebuttal-7D65/3435_NT_T1_barplot.png), [3435_NT_T1 Table](https://anonymous.4open.science/r/celltreeqm-rebuttal-7D65/3435_NT_T1_table.png), [3435_NT_T6 Barplot](https://anonymous.4open.science/r/celltreeqm-rebuttal-7D65/3435_NT_T6_barplot.png), [3435_NT_T6 Table](https://anonymous.4open.science/r/celltreeqm-rebuttal-7D65/3435_NT_T6_table.png).
>
> As in the C. elegans datasets, we show that we can achieve significant learning and CellTreeQM has better results, especially generalizing to unknown quartets than other baseline approaches. We note that due to the limited time for this response, we have not been able to consider tuning the model for these datasets and we expect better performance might be achieved with additional work.
>
> Please see FYFt Concern 2 for single cell trajectory methods and FYFt Concern 3 for phylogeny methods.

---

> > ### Comment · Reviewer_vktx · 2025-04-03
> >
> > Upgraded my score in light of more experiments.

---

### Official Review · Reviewer_RR5y · 2025-03-13

**Overall Recommendation:** 5

**Summary:**

This paper poses the reconstruction of lineage trees from phenotypic data as a metric learning problem, and devises a contrastive loss function to learn a metric given partial information about the topology of the lineage tree. The authors test this algorithm on ground truth data from C. elegans, as well as simulated data of branching random processes, and find that it performs quite well relative to reasonable baselines.

**Claims And Evidence:**

I think the most important claim of this paper is that the presented problem is indeed an important one, and on this I wholeheartedly agree. The cited literature is thorough and wonderfully bridges the biological motivation with the computational problem at hand. Phenotypic information is widely available, yet it is important to have methods such as these that bridge those with more targeted approaches and prior knowledge about cell lineage. I present this in the claims section rather than the relation to broader literature because it is a key source of originality in this paper to identify and tackle a problem that has not been addressed by several machine, learning papers already.

there are many more specific claims made in the paper, and for the most part, these are well supported by the experiments presented. the supervised and semi supervised settings are particularly convincing.

I think the one overstated claim in this paper is that this method is useful in unsupervised settings. well the authors do remark that this is preliminary, it is still mentioned in the abstract as contribution of this paper. However, the unsupervised setting is not thoroughly evaluated, and little attempt is made to compare with a large number of hierarchical clustering approaches that are widely used in single cell analysis. Nonetheless, I do not think the unsupervised setting is the main contribution of this paper.

**Essential References Not Discussed:**

the literature is very thorough already. I might suggest papers to further augment the case that phenotypic information alone is insufficient to reconstruct the lineage, as in the purely unsupervised case. https://www.nature.com/articles/s41586-021-04237-0 presents one example.

**Experimental Designs Or Analyses:**

the experimental designs are sound.

**Methods And Evaluation Criteria:**

the combination of the branching diffusion process simulation with ground truth lineage information in flatworms is nice to see. I furthermore appreciate the choice of quantitative evaluations on these two data sets. these are well suited to the problem at hand.

**Other Comments Or Suggestions:**

line 340: “known quartetes” -> “known quartets”
line 316: "we training an model" -> "we train a model"

**Other Strengths And Weaknesses:**

the principal strength of this work is in its ideation of a novel research direction for a real problem in developmental biology, presented by the available methods and the perhaps surprising to similarity between phenotype and lineage in single cell data. The author is apply interesting machine, learning ideas in novel ways to address this problem.

I cannot identify many weaknesses in this paper. The ability of this method to work in purely unsupervised settings is somewhat untested, and in the setting for which other methods exist these are not compared.

**Questions For Authors:**

I'm curious about the application of this method to settings in which we know the cell type lineage tree, but in a particular dataset may have millions of cells. This is a example of partial information about the cellular tree. With this method still be applicable.

**Relation To Broader Scientific Literature:**

The construction of lineage trees is an important problem in developmental biology, yet it is rare to have experimental validated trees down to the cellular level. Thus, there is a real need for papers that can augment limited experimental data. I particularly appreciate the starting point of this manuscript, which recognizes that phenotypic information does not correlate well in certain cases with lineage. thus, this paper targets an important problem in biology with novel methods and ideas with in machine learning, pioneering an important path forward for both fields.

**Theoretical Claims:**

no novel theoretical claims are made.

---

> ### Author Rebuttal · Authors · 2025-03-31
>
> ## Concern 1: Overstated claim about unsupervised learning
> > Although acknowledged as preliminary, unsupervised performance is mentioned in the abstract.
> ### Response
>
> We appreciate the comment concerning the limited investigation of the unsupervised setting. We demonstrate the unsupervised setting with the brownian motion dataset and we are capable of filtering out independent noise during training, demonstrated by the dash lines in Figure 5. We agree that the investigation under unsupervised setting is still preliminary so we have reworded the abstract and other parts of the manuscript describing our unsupervised results.
>
> **In Abstract:** “we systematically explore, supervised, weakly supervised, and unsupervised…” → “we systematically explore weakly supervised training settings at different levels of information..”
>
> **Section 7.4:**
>
> From:
>
> “Although the problem is more difficult—and in practice, purely unsupervised lineage inference
> from phenotypes alone can be error-prone—we find that CellTreeQM can still learn a representation that partially respects tree-metric properties. See details in §F.5.
>
> To:
>
> “Here, we used a data-driven estimate of quartet order in the latent space. CellTreeQM can still learn a representation that partially respects tree-metric properties in the limited setting of the simulation data but performance on real data was not significant, suggesting better strategies are needed. See details in §F.5.”
>
> **Discussion section:**
>
> From:
>
> “Empirical results in supervised, weakly supervised, and unsupervised settings show that CellTreeQM considerably improves…”
>
> To:
>
> Empirical results in supervised and weakly supervised settings show that CellTreeQM considerably improves…”
>
> ## Concern 2: Comparing to clustering methods
> > Not thoroughly evaluated or compared to clustering methods widely used in single-cell analysis.
>
> ### Response
> As noted by the reviewer, we did not compare to various unsupervised clustering methods widely used in single cell literature. One of the key reasons is that the clustering methods, as currently used, are applied to whole transcriptomes. Carefully examined biological processes such as in C. elegans, show that phenotypic types are not the same as cell-lineage clades (also, we appreciate all the reviewers mentioning this fundamental concept in their comments).
>
> In fact, this disjunction is the central problem in cell lineage estimation. While we have not systematically studied clustering methods, we have now added examples of applying cell trajectory methods–we note that these methods work in the original feature space and attempts to create a piecewise one-dimensional (i.e., tree-graph) parameterization of the full phenotypic state, rather than trying to extract lineage-specific information. Please see our response to ekW4 Concern 1 and FYFt Concern 2.
>
> ## Concern 3: Missing literature
> > I might suggest papers to further augment the case that phenotypic information alone is insufficient to reconstruct the lineage, as in the purely unsupervised case. https://www.nature.com/articles/s41586-021-04237-0 presents one example.
> ### Response
> We thank the reviewer for this paper and we added this reference and revised the Key Challenges section to better reflect this comment.
>
> ## Concern 4: Scalability
> > I'm curious about the application of this method to settings in which we know the cell type lineage tree, but in a particular dataset may have millions of cells. This is a example of partial information about the cellular tree. With this method still be applicable.
> ### Response
> Thank you for this interesting question. We believe there are several approaches and barriers. First, as discussed above, typical cell types are not monophyletic, therefore cell type lineage tree is likely to be a mixture of cell tree lineages. As shown in [this figure](https://anonymous.4open.science/api/repo/celltreeqm-rebuttal-7D65/file/cell_type-vs-cell_lineage.png?v=50c079c2).
>
> Nevertheless, we could use its hierarchy as prior knowledge to designate known quartets. Ideally, the latent space with additive tree metric should be universal and therefore learnable with subsets of the millions of cells–perhaps using the strategy of random mini-batches. The key barrier will be the application of phylogeny algorithms. Even the heuristic algorithms are O(n^2)~O(n^3), which might be practically impossible for millions. Currently, large-scale phylogenies such as viral isolate phylogenies are built using a mixture of inferring “backbone” trees and using insertional type of approaches. We hope to continue our studies to include extreme scale up problems.

---

### Decision · Program_Chairs · 2025-05-01

**Decision:**

Accept (poster)

**Comment:**

The paper proposes an approach reconstruct cell lineage relationships from phenotypic data. The main contribution is a new loss function that formulates lineage reconstruction as tree-metric learning problem and enables supervised and weakly supervised learning. The reviewers are generally supportive of the paper but criticized the limited empirical evaluation as well as the potentially overlooked link to trajectory inference methods. During the rebuttal period, the authors could show that the method can be applied successfully on additional datasets and clarified how their problem setting differs from that tackled by trajectory inference methods. As a result, all reviewers that engaged in the discussion are now in support of the paper. Hence, I recommend accepting it.